



# Extended seasonal prediction of Antarctic sea ice using ANTSIC-UNet

Ziying Yang[1, 2], Jiping Liu[3], Mirong Song[1], Yongyun Hu[4], Qinghua Yang[3], Ke Fan[3]

[1]Institute of Atmospheric Physics, Chinese Academy of Sciences, Beijing 100029, China
[2]University of Chinese Academy of Sciences, Beijing 101408, China
[3]School of Atmospheric Sciences, Sun Yat-sen University, and Southern Marine Science and Engineering Guangdong Laboratory (Zhuhai), Zhuhai 519082, China
[4]Department of Atmospheric and Oceanic Sciences, School of Physics, Peking University, Beijing 100087, China

*Correspondence to*: Jiping Liu (liujp63@mail.sysu.edu.cn)

**Abstract.** Antarctic sea ice has experienced rapid change in recent years, which garners increasing attention for its prediction.
In this study, we develop a deep learning model (named ANTSIC-UNet) trained by physically enriched climate variables and evaluate its skill for extended seasonal prediction of Antarctic sea ice concentration (up to 6 months in advance). We compare the predictive skill of ANTSIC-UNet in the Pan- and regional Antarctic with two benchmark models (linear trend and anomaly persistence models). In terms of root-mean-square error (RMSE) for sea ice concentration and integrated ice-edge error (IIEE), ANTSIC-UNet shows much better skills for the extended seasonal prediction, especially for the extreme events in recent years,
relative to the two benchmark models. The predictive skill of ANTSIC-Unet is season and region dependent. Low values of RMSE are found from autumn to spring in the Pan-Antarctic and all sub-regions for all lead times, but large values of RMSE are found in summer for most sub-regions which increase as lead times increase. Small values of IIEE are found in summer at 1-3 month lead, large errors occur from November to January as the lead time exceeds 2-4 months. The Pacific and Indian Oceans show better predictive skills at the sea ice edge zone in summer compared to other regions. Moreover, ANTSIC-UNet
shows good predictive skill in capturing the interannual variability of Pan-Antarctic and regional sea ice extent anomalies. We also quantify variable importance through a post-hoc interpretation method. It suggests in addition to sea ice conditions, the ANTSIC-UNet prediction at short lead times shows sensitivity to sea surface temperature, radiative flux, and atmospheric circulation. At longer lead times, zonal wind in the stratosphere appears to be an important influencing factor for the prediction.

## 1 Introduction

Antarctic sea ice is an essential component of the climate system, which strongly affects the local atmosphere and ocean and the extrapolar Southern Hemisphere through dynamic and thermodynamic processes, particularly in a warming climate (Massom and Stammerjohn, 2010; Kidston et al., 2011; Abernathey et al., 2016; Zhu et al., 2023). The total Antarctic sea ice extent (SIE) has gradually increased until 2014 since the late 1970s and then abruptly decreased (Turner et al., 2013; Hobbs et al., 2016; Comiso et al., 2017; Fogt et al., 2022; Liu et al., 2023). Antarctic sea ice shows large seasonal and interannual
variability, and its trend is spatially heterogeneous (Liu et al., 2004; Raphael and Hobbs, 2014; Libera et al., 2022).



Compared to the Arctic, the prediction of Antarctic sea ice has received much less attention, but it is also in demand associated with the increase in planning operational activities like scientific research, tourism, and fishing in the Southern Ocean (Bushuk et al., 2021; Libera et al., 2022). Statistical models, such as the Markov model (e.g., Chen and Yuan, 2004; Pei, 2021) and the Koopman mode decomposition model (Hogg et al., 2020), have been employed to forecast seasonal Antarctic sea ice concentration. In general, these statistical models were inferior to the anomaly persistence model for some seasons and regions. There have been limited efforts to forecast seasonal Antarctic sea ice using dynamic models due to the challenges associated with faithfully simulating complex air-ice-sea interaction processes in the Southern Ocean (Morioka et al., 2019; Bushuk et al., 2021). Most dynamical forecast systems overestimate the extent of the Antarctic sea ice edge at the sub-seasonal scale with their predictive skill falling below climatological benchmarks (Zampieri et al., 2019). Starting in 2017, the Sea Ice Prediction Network South (SIPN South) has coordinated the evaluation of forecasting methods and systems used to predict summer Antarctic sea ice (Massonnet et al., 2023). The evaluation reveals that both statistical and dynamical models have substantial bias and ensemble spread.

Deep learning (DL) method has been widely used for Arctic sea ice prediction at various temporal scales (e.g., Chi and Kim, 2017; Kim et al., 2020; Y. Ren and X. Li, 2021). Andersson et al. (2021) introduced IceNet to predict probabilities of Arctic sea ice edge with uncertainty quantification. Y. Ren and X. Li (2023) developed a DL method with a physically constrained loss function to improve Arctic sea ice predictions at lead times of 90 days. In contrast, very limited effort has been made to apply DL method to Antarctic sea ice prediction and associated assessments are still at an early stage. For the SIPN South summer Antarctic sea ice extent forecast (Massonnet et al., 2023), one contributor provided the prediction using a k-nearest neighbors (KNN) method. Recently, Wang et al. (2023) developed a SIPNet model with encoder-decoder structure for subseasonal Antarctic sea ice concentration prediction, which outperforms some dynamic models and advanced linear statistical models. Both the DL methods were trained by pure historical sea ice concentration data without considering underlying physical processes governing the variation of Antarctic sea ice.

The purposes of this study are to 1) develop a DL model, named ANTSIC-UNet, to achieve extended seasonal prediction of Antarctic sea ice concentration by considering not only sea ice itself but also a wealth of knowledge in terms of ocean-ice-atmosphere interactions, 2) assess the predictive skill of ANTSIC-UNet for both Pan- and regional Antarctic sea ice, especially recent extreme years, and 3) conduct a post-hoc interpretation method to quantify the variable importance that affects sea ice predictability.

## 2 Data and Method

### 2.1 Data

In this study, monthly Antarctic sea ice concentration (SIC) data obtained from the National Snow and Ice Data Center (NSIDC) (https://nsidc.org/data/nsidc-0079/versions/3) is used as the input of ANTSIC-UNet, which derived from brightness temperature of the Scanning Multichannel Microwave Radiometer (SMMR), the Special Sensor Microwave/Imager (SSM/I)



sensors, and the Special Sensor Microwave Imager/Sounder (SSMIS). The SIC data has a size of $316 \times 332$ with a spatial resolution of 25km, spanning from 1979 to 2023. We also use the linear trend prediction of SIC as the input which is computed
by the linear least squares fitting for the calendar month corresponding to the period of 1-year ahead from the target month. The atmospheric and oceanic variables obtained from the ECWMF Reanalysis v5 (ERA5, Hersbach et al., 2020) and Ocean Reanalysis System 5 (ORAS5, Zuo et al., 2019) are also used as the inputs, which are related to dynamic and thermodynamic processes of Antarctic sea ice. These variables include 2m air temperature (T2), 500-hPa air temperature (T500), sea surface temperature (SST), ocean temperature (PT), ocean heat content for the upper 300m (OHC300), downwelling solar radiation
(DSR), upwelling solar radiation (USR), sea level pressure (SLP), 500-hPa geopotential height (H500), 250-hPa geopotential height (H250), 10m u-component of wind (U10), 10m v-component of wind (V10), and 10-hPa zonal wind (U10hPa). The averaged ocean temperature at different depths in the upper Southern Ocean, 50-100m (PT50) and 100-150m (PT100), has been calculated. Before integrating into ANTSIC-UNet, these variables are bilinearly interpolated to the NSIDC sea ice polar stereographic grid and standardized. Additionally, a land mask obtained from the NSIDC is used for the consistency of SIC
and other variables.

The input vector is a 3-dimensional matrix with the size of $316 \times 332 \times 57$. 57 is the dimension of the variables, including sea ice concentration for the past 12 months, the linear trend prediction of sea ice concentration for the following 6 months, 14 climate variables for the past 1 to 3 months, and the land mask. All variable fields are mapped on $316 \times 332$ grids (see Table 1 for the details of all input variables). The final output provides the 6-month forecast of Antarctic sea ice concentration.

## 80   2.2 ANTSIC-UNet model

In this study, we construct an ensemble deep learning model, aiming at the extended seasonal Antarctic sea ice concentration prediction. The ANTSIC-UNet consists of 20 members possessing the encoder and decoder structure associated with a fully convolutional network (Fig. 1). Such encoder and decoder framework is also employed in IceNet used for Arctic probabilistic forecasting (Andersson et al., 2021), and originally designed in the U-Net for image recognition (Ronneberger et al., 2015).
The encoder is designed to extract abstract features through convolutional layers and downscale features using maxpooling layers, which increases the robustness and reduces the amount of computation for a deeper network. The decoder is designed to recover and reconstruct the abstract features through convolutional layers, and generate outputs of the same spatial size as the inputs through unsampling layers. Four skip connections linking feature maps in the same semantic level provide multi-scale and multi-level information and retain high-resolution details in the initial convolution process. To avoid deformation,
we resize the spatial shape to $336 \times 320$ by the nearest neighbor method before the encoder and adopt a padding technique to avoid too much data reduction. Finally, we extract the slices from the output module which contains six convolutional layers using the sigmoid activation function to transform output values.

We divide the data into three groups: the training data from 1979 to 2011, the validation data from 2012 to 2019 (with exclusion years 2014 and 2017), and testing data in 2017, from 2020 to 2023 (anomalously low extent period) and 2014 (record high)
for independent evaluation. An early stopping strategy is adopted to avoid overfitting when the performance on the validation



data does not improve after 10 epochs as suggested by Prechelt (2012). The testing data do not participate in the training process so that the performance of the testing data further estimates the generalization of ANTSIC-UNet for adapting new data.



**Figure 1. Configuration of ANTSIC-UNet model used for extended seasonal Antarctic sea ice prediction.**



| Input variables | Variable long name | Source | Lead or lag (months) |
|---|---|---|---|
| SIC | sea ice concentration | NSIDC | 1 to 12 |
| SIC trend | linear trend forecast for sea ice concentration | NSIDC | 1 to 6 |
| T2A | 2 m air temperature anomaly | ERA5 | 1 to 3 |
| T500A | 500-hPa air temperature anomaly | ERA5 | 1 to 3 |
| SSTA | sea surface temperature anomaly | ERA5 | 1 to 3 |
| PT50A | ocean temperature anomaly averaged over 50-100 m | ORAS5 | 1 to 3 |
| PT100A | ocean temperature anomaly averaged over 100-150m | ORAS5 | 1 to 3 |
| OHC300A | ocean heat content anomaly for the upper 300 m | ORAS5 | 1 to 3 |
| DSRA | surface downward solar radiation | ERA5 | 1 to 3 |
| USRA | surface upward solar radiation | ERA5 | 1 to 3 |
| SLPA | sea level pressure anomaly | ERA5 | 1 to 3 |
| H500A | 500-hPa geopotential height anomaly | ERA5 | 1 to 3 |
| H250A | 250-hPa geopotential height anomaly | ERA5 | 1 to 3 |
| U10hPa | 10-hPa zonal wind | ERA5 | 1 to 3 |
| U10 | 10 m zonal wind | ERA5 | 1 |
| V10 | 10 m meridional wind | ERA5 | 1 |
| landmask | Southern Hemisphere land mask | NSIDC | N/A |

**Table 1. The information of all input variables for ANTSIC-UNet**

**2.3 Evaluation metrics**

In this study, the linear trend and anomaly persistence predictions are used as benchmarks to assess the predictive skill of ANTSIC-UNet. The linear trend prediction is described in section 2.1. The anomaly persistence prediction is calculated as follows:

$$SIC_{pred}(t,\tau) = SIC_{clim}(t) + SIC_{anom}(0,\tau) \tag{1}$$

where $SIC_{pred}$ is the target month ice concentration at the lead time $\tau$, $SIC_{clim}(t)$ is the climatogy ice concentration at the target month, and $SIC_{anom}(0,\tau)$ is the observed ice concentation anomaly at the initial time. The climatology for each month is computed for the period of the training data (1979-2011).

We quantify the predictive skill of both the Pan- and regional Antarctic sea ice using four metrics: 1) root-mean-square error (RMSE), 2) anomaly correlation coefficient (ACC), 3) mean squared error skill score (MSSS), and 4) integrated ice-edge error (IIEE). RMSE reflects the proximity between the prediction and observation. ACC is a measure of the accuracy of the prediction model based on the relationship between the predicted and observed deviation from the climatology (Wang et al.,



2016). MSSS is a skill score based on a comparison of the prediction and climatology that takes into account both ACC and conditional bias. The value of MSSS varies from -1 to 1, with a negative value indicating no predictive skill and 1 indicating

a perfect forecast (Murphy, 1988). For skillful prediction, here we use ACC = 0.5 and MSSS = 0.0 as the lowest limit, which are widely used in previous research (e.g., Goddard et al., 2012; Choi et al., 2016; Bushuk et al., 2021). The integrated ice-edge error (IIEE) is a verification metric for sea ice forecasts representing the sum of overestimated and underestimated sea ice extent where sea ice concentration > 15% (Goessling et al., 2016). These metrics are calculated as follows:

$$RMSE = \sqrt{MSE} = \sqrt{mean(\sum(p - o)^2)}, \tag{2}$$

$$ACC = \frac{\sum(p - \bar{p})(o - \bar{o})}{\sqrt{\sum(p - \bar{p})^2}\sqrt{\sum(o - \bar{o})^2}}, \tag{3}$$

$$MSSS = 1 - \frac{MSE_{pred}}{MSE_{clim}} = 1 - \frac{mean(\sum(p - o)^2)}{mean(\sum(\bar{o} - o)^2)}, \tag{4}$$

$$IIEE = SIE_p \cup SIE_o - SIE_p \cap SIE_o, \tag{5}$$

where $p$ is the predicted ice concentration or sea ice extent by ANTSIC-UNet and $o$ is the observed ice concentration or ice extent; $\bar{p}$ and $\bar{o}$ are the mean of the prediction and observation.

## 2.4 Variable importance analysis

We use the permutation feature importance approach to determine which variables are important for Antarctic sea ice prediction in ANTSIC-UNet. This method was introduced by Breiman (2001) and Fisher et al. (2018) to interpret the model's decisions. Specifically, when a particular variable is selected, the original input feature matrix is $\boldsymbol{X}_{orig}$ and the permutation feature matrix is $\boldsymbol{X}_{perm}$. The evaluation metric $e_{i,j}$ used is the root-mean-square error (RMSE) between the output $f_{i,j}$ (the

predicted SIC by the trained model for the target month at the lead time ranging from 1 to 6 months) and the target $\boldsymbol{Y}_i$ (observed SIC) for a given month. Thus, the feature importance value $FI_{i,j}$ is defined as the accuracy change of the evaluation metric where i refers to the target month to be predicted and j refers to the lead month.

$$FI_{i,j} = e_{i,j}{}^{perm} - e_{i,j}{}^{orig}, \tag{6}$$

where

$$e_{i,j}{}^{orig} = RMSE(\boldsymbol{Y}_i; f_{i,j}(\boldsymbol{X}_{orig})) = \sqrt{mean(\sum(\boldsymbol{Y}_i - f_{i,j}(\boldsymbol{X}_{orig}))^2)}, \tag{7}$$

$$e_{i,j}{}^{perm} = RMSE\left(\boldsymbol{Y}_i; f_{i,j}(\boldsymbol{X}_{perm})\right) = \sqrt{mean(\sum(\boldsymbol{Y}_i - f_{i,j}(\boldsymbol{X}_{perm}))^2)}, \tag{8}$$

The importance of each particular variable is measured by 1) randomly shuffling the variable across spatial grids and replacing it in the original input vector to generate a new input vector, and 2) calculating the error of the evaluation metric after permuting the variable. The positive increase of $FI_{i,j}$ means that the variable is important. By contrast, no change and decrease of $FI_{i,j}$



indicates that the variable has little role. Here we iteratively the permutation and evaluation for each input variable and repeat the procedure 10 times. The mean feature importance value is calculated with the testing data for the period of 2020-2023.

## 3 Results

### 3.1 Pan-Antarctic and regional predictive skill

Pan-Antarctic sea ice concentration predictions from ANTSIC-UNet, linear trend and anomaly persistence models for the
testing years averaged for all lead times are shown in Table 2. Overall, ANTSIC-UNet has smaller RMSEs and significantly reduced IIEE compared to the linear trend and anomaly persistence models. Considering the metrics vary with lead times and different regions, we compare the three models for lead times ranging from 1 to 6 months for the Pan-Antarctic and five sub-regions (Fig. 2). For ANTSIC-UNet and anomaly persistence model, both RMSE and IIEE grow with increasing lead time, reflecting a decrease of predictive skill for the extended seasonal forecast. Compared to the anomaly persistence model,
ANTSIC-UNet exhibits significantly lower RMSE over the entire Antarctic and all sub-regions for all lead times, except for the Indian Ocean as the lead time exceeds 3 months. RMSE of ANTSIC-UNet also exceeds the linear trend model as the lead time exceeds 3 months, which is due to the reduced predictive skill in the Indian Ocean, Pacific Ocean, Amundsen and Bellingshausen Seas. Encouragingly, the IIEE of ANTSIC-UNet is consistently smaller than that of the two benchmark models, though it is comparable to the linear trend model as lead times exceed 3 months in the Amundsen and Bellingshausen Seas.
Overall, ANTSIC-UNet shows high predictive skill in the Weddell and Ross Seas, which outperforms the two benchmark models.

|  | ANTSIC-UNet | Linear trend | Anomaly persistence |
|---|---|---|---|
| RMSE | 0.21 | 0.22 | 0.23 |
| IIEE | 1.68 | 2.13 | 2.47 |

**Table 2. The averaged predictive skill of Antarctic sea ice for ANTSIC-Unet, linear trend and anomaly persistence**
**models for all testing years.**



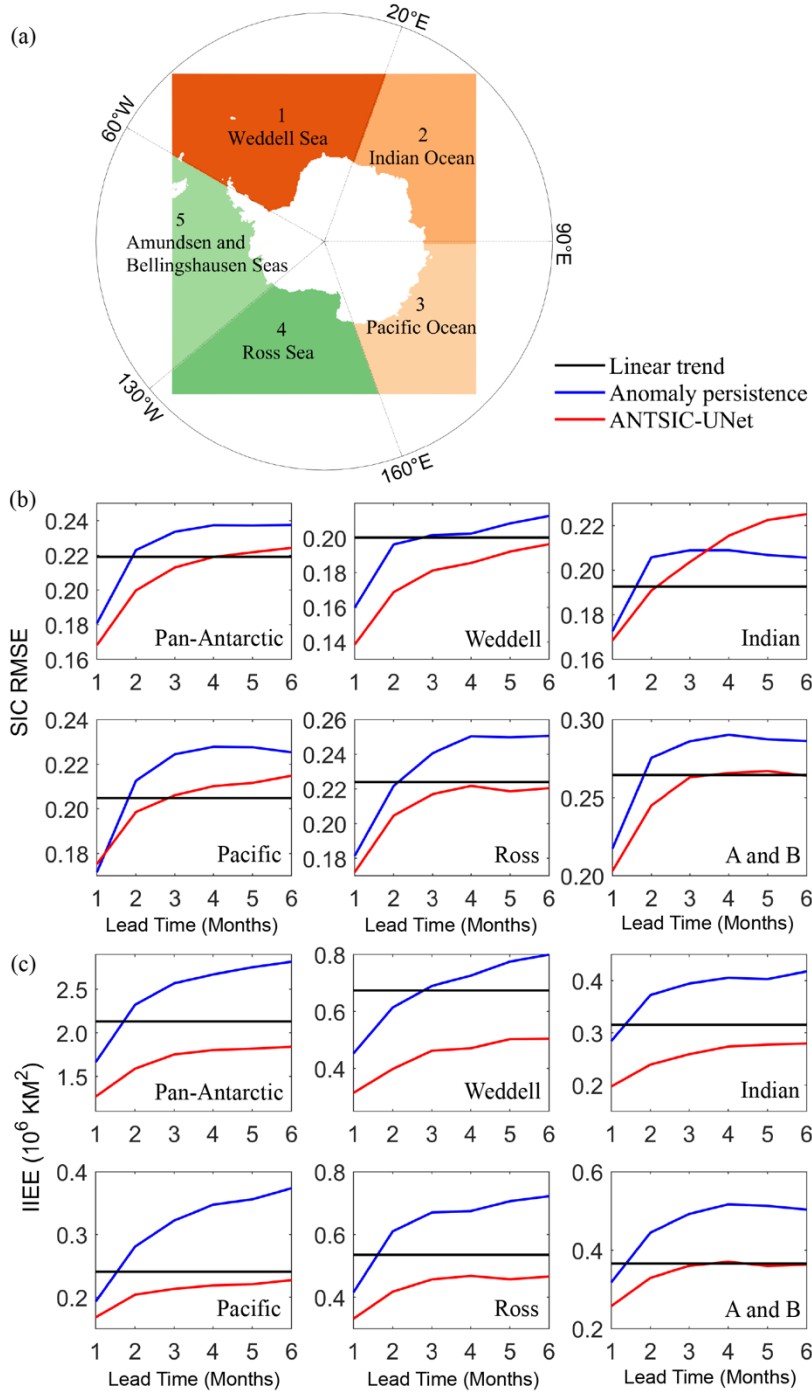

**Figure 2. (a) Domian of sub-regions: 60°W–20°E (Weddell Sea), 20°–90°E (Indian Ocean), 90°–160°E (Pacific Ocean), 160°E–130°W (Ross Sea), and 130°–60°W (Amundsen and Bellingshausen Seas). (b) and (c) the averaged predictive skill of Pan- and regional Antarctic sea ice for ANTSIC-UNet, linear trend and anomaly persistence predictions. (b) SIC RMSE and (c) IIEE.**



Fig. 3 shows the spatial distribution of February and September SIC. In February (seasonal minimum), the linear trend model overestimates SIC in the Ross Sea and western and central Weddell Sea and underestimates SIC in the Amundsen and Bellingshausen Seas. Compared to the linear trend model, the anomaly persistence model has relatively small biases at 1-
month lead. However, the magnitude and coverage of the biases become larger as the lead time increases, i.e., it shows large positive (negative) biases in parts of the eastern Pacific sector (the Indian sector) at 5-month lead. Moreover, the anomaly persistence model leads to the fake northward expansion of the bias due to sea ice during the initial months (i.e., spring) having broader coverage than the target month (i.e., summer) as the lead time increases. By contrast, the ANTSIC-UNet prediction shows the smallest biases (mostly negative in much of the Antarctic) at 1-month lead. As the lead time increases, the magnitude
of the biases gradually increases, except that the negative bias in the Ross Sea changes to the positive. In September (seasonal maximum), the linear trend and anomaly persistence (at 1-month lead) models tend to have alternating negative and positive biases near the sea ice edge. By contrast, the ANTSIC-UNet prediction has smaller and mostly negative biases for much of the Antarctic at 1-month lead. As the lead time increases, both the ANTSIC-UNet and anomaly persistence models show biases become larger in the sea ice edge zone. Moreover, large biases also appear in the compact ice zone for the anomaly persistence
model.

Fig. 4 shows spatially and temporally averaged RMSE and IIEE between the ANTSIC-UNet predictions and observations for each target month and different lead times. In terms of RMSE, Pan-Antarctic exhibits low values from autumn to spring (from April to November), though there is an increase in RMSE during summer months (from December to March) as the lead time exceeds 2 months. In terms of IIEE, Pan-Antarctic has small values at 1-month lead, which extend to 2-3 month lead in
February and March. In general, the values of IIEE increase as lead times increase, and large values occur from November to January as the lead time exceeds 2-3 months. As shown in Fig. 4b1-f1, the large values of RMSE are also found in summer for all sub-regions, but relatively small values are found in the Weddell Sea. For IIEE in Fig. 4b2-f2, all sub-regions show similar distributions, except that the low IIEE in the Indian and Pacific Oceans have broader coverage. Increased IIEEs are found in the Weddell Sea (Ross Sea) from November to January (from December to March) as the lead time exceeds 2-3
months. Overall, the Pacific and Indian Oceans show better predictive skills at the sea ice edge zone in summer relative to other regions.







**Figure 3.** The monthly mean sea ice concentration of the NSIDC observations for (a) February and (e) September, and the errors predicted by ANTSIC-UNet (b1-b3, f1-f3), the linear trend model (c and g), and anomaly persistence model (d1-d3, h1-h3) at lead time of 1, 3, and 5 months for February (upper panel) and September (lower panel) during the testing years.




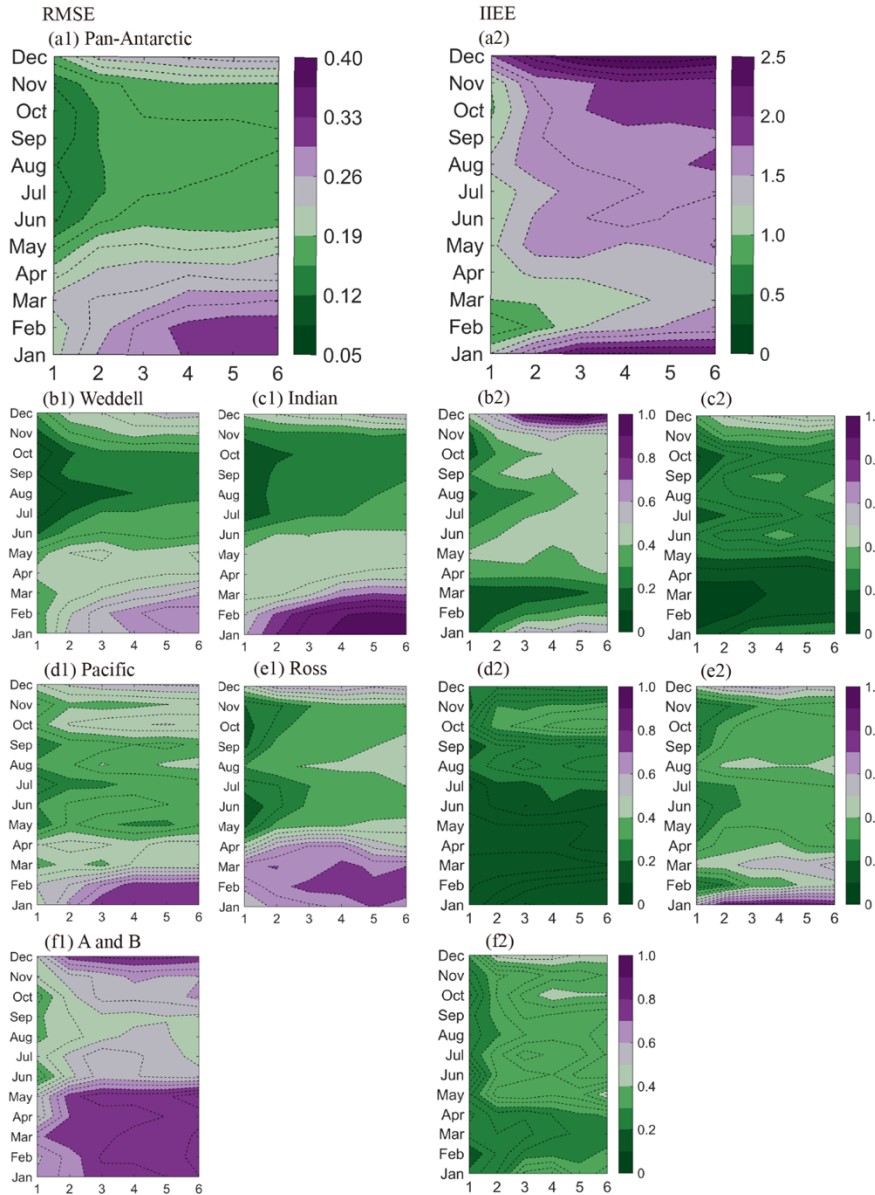

**Figure 4. The predictive skill of sea ice concentration (spatially and temporally averaged during the testing years) in terms of RMSE and IIEE (units: million square kilometers) between the ANTSIC-UNet predictions and NSIDC observations for different target months and forecast lead times.**

**3.2 Predictive skill for interannual variability**

We assess the performance of the predicted year-to-year variability of Pan-Antarctic and regional sea ice extent (SIE) anomalies (Fig. 5). For the Pan-Antarctic, the observed ice extent anomaly shifts from the positive phase to the negative phase around 2016 (Fig. 5a). Both the linear trend and anomaly persistence models cannot capture the observed shift after 2016, and



the anomaly persistence model shows much larger positive anomalies and variability compared to the observation. By contrast,

ANTSIC-UNet reproduces the observed shift during 2014-2017 and the predicted interannual variability is well correlated with the observation (R=0.76). Moreover, the majority of the observed ice extent anomalies fall within the spread of the ANTSIC-UNet prediction, which is also true for most sub-regions (Fig. 5b-f). The highest correlation is found in the Weddell Sea (R=0.79), followed by the Indian Ocean (R=0.63) and Ross Sea (R=0.59). The Pacific Ocean, Amundsen and Bellingshausen Seas have relatively low correlations. Thus ANTSIC-Unet outperforms two benchmark models from the

perspective of the SIE interannual variability prediction.

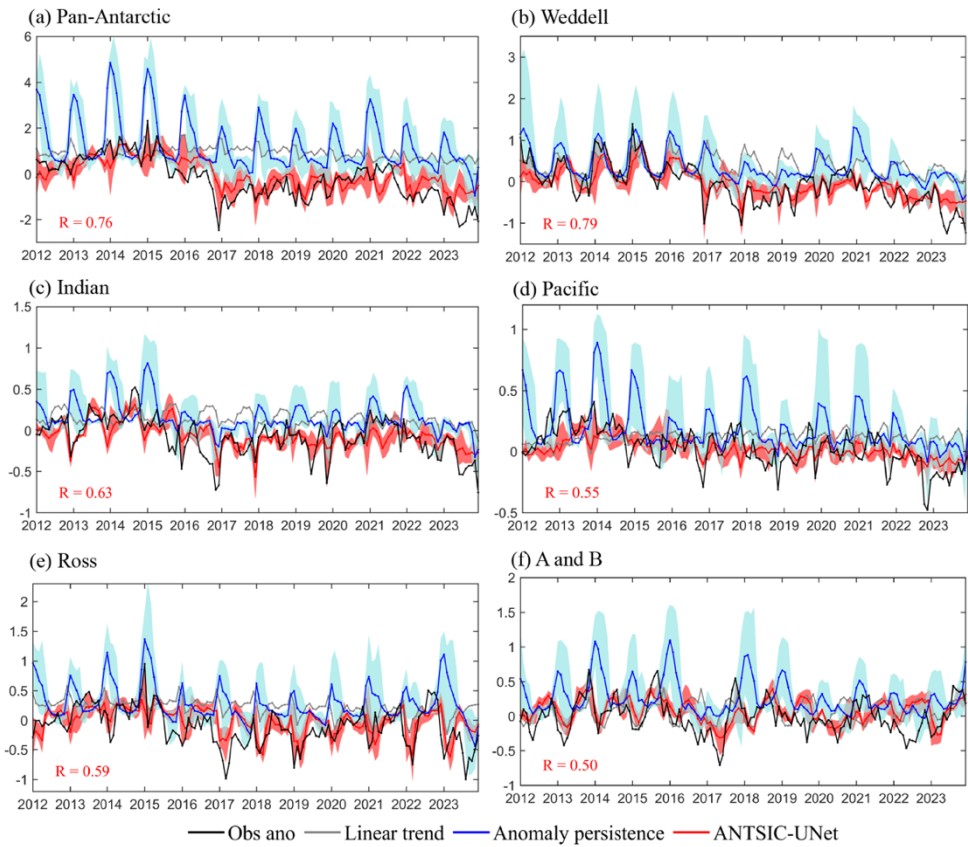

**Figure 5. Time series of sea ice extent anomalies from 2012 to 2023 (validation and testing years) for Pan- and regional Antarctic. The NSIDC observations, the linear trend model, the anomaly persistence and ANTSIC-UNet model lead time averaged predictions are represented by black, grey, blue and red lines. The red (blue) shading represents the ensemble spread of ANTSIC-Unet (anomaly**

**persistence model) at different lead times. (units: million square kilometers)**

Fig. 6 further shows the evaluation metrics (ACC and MSSS) between the observed and predicted interannual sea ice extent. For the Pan-Antarctic, high values of ACC are found from January to September at 1-3 months lead, which decrease as the lead times increase (Fig. 6a). Reduced values of ACC are found from October to December as the lead time exceeds 2 months. MSSS exhibits a similar pattern as that of ACC (Fig. 6b). All sub-regions show similar distributions, high values of ACC and

MSSS at 1-month lead and slowly decreasing with increasing lead times. Low values of ACC and MSSS occur in the Indian



Ocean from Januray to March, the Pacific Ocean from November to January, and the Amundsen and Bellingshausen Seas from September to October, which limit the interannual predictive skill of the Pan-Antarctic. Overall, the Weddell and Ross Seas have broad coverage of high ACC and MSSS which suggests the possibility of long-lead extended seasonal predictions there.

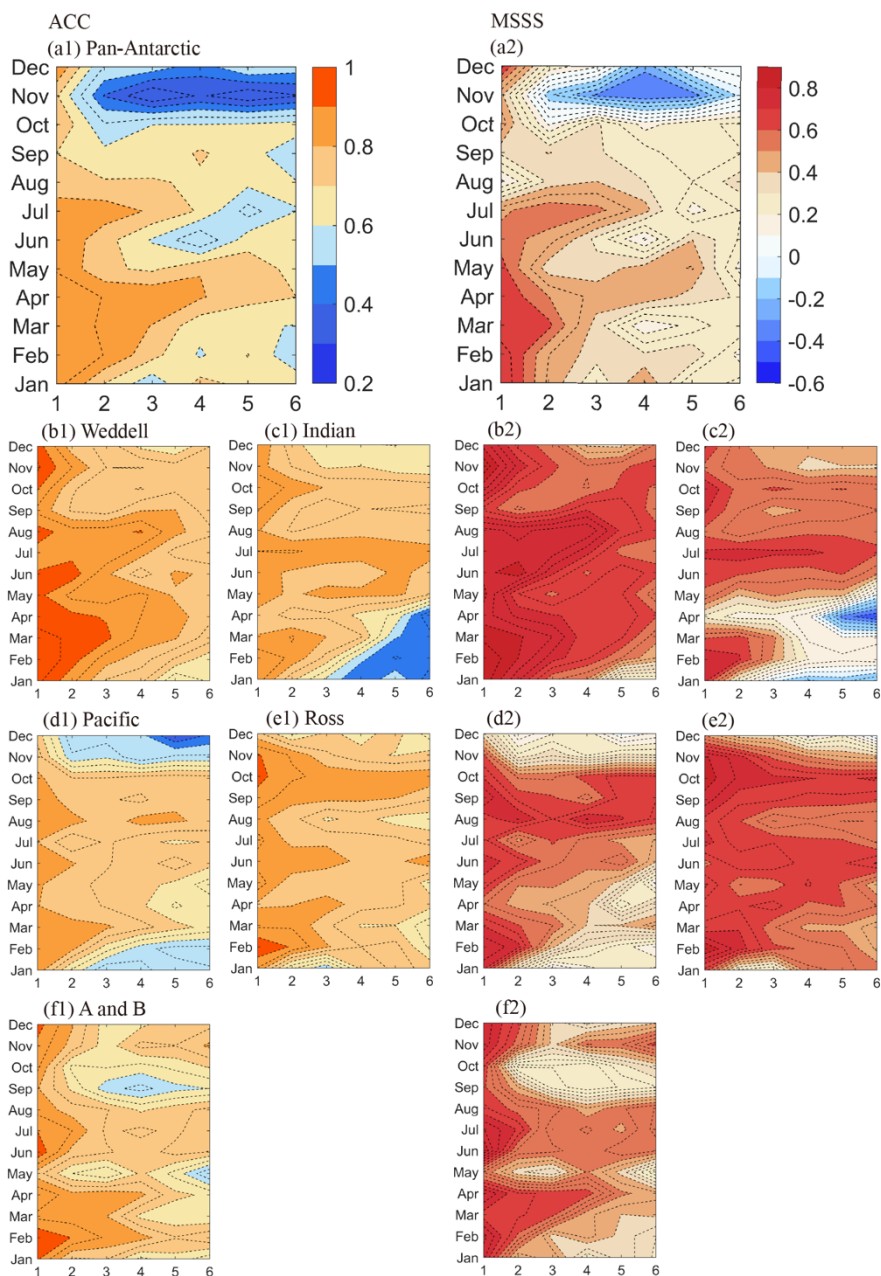

**Figure 6. The ACC (a1-f1) and MSSS (a2-f2) between the observed and ANTSIC-UNet predicted regional SIE anomalies for different target months and forecast lead times during 1981-2023.**



### 3.3 Extremes cases

Next, we evaluate to what extent the ANTSIC-UNet prediction can capture extreme years. The average predictive skills for the three extremely low sea ice extent years averaged for all lead times are shown in Table 3. During all extreme years, 235 ANTSIC-UNet exhibits the smallest RMSEs and improves sea ice edge predictions with notably reduced IIEE, compared to the linear trend and anomaly persistence models. The spatial distribution of February and September SIC of 2023 (record low) is shown in Fig. 7. In February, the linear trend model overestimates sea ice concentration for much of the Antarctic. The anomaly persistence model shows clusters of large positive biases near the coastal area and extended northward coverage of negative biases at 1-month lead, and both magnitude and coverage of the biases increase dramatically as the lead time increases. 240 ANTSIC-UNet exhibits better performance than the two baseline models with smaller sea ice edge error for all lead times, though as lead time increases, the positive biases in the Amundsen and Ross Seas gradually increase. In September, the ANTSIC-UNet prediction shows smaller biases in the entire Antarcic at 1-month lead compared to the two benchmark models, and still outperforms the two models in most regions as the lead time increases. Though there are different spatial distributions of SIC errors for 2017 and 2022, ANTSIC-UNet also shows superior predictive skill (Figs. S1 and S2). 245 The predictive skill of seasonality errors of extremely low sea ice extent of 2023 based on ANTSIC-UNet and two benchmark models are further accessed against the NSIDC observations (Fig. 8). Both the linear trend and anomaly persistence prediction models excessively overestimate the SIE in the Pan-Antarctic and all sub-regions for nearly all months, except for the Amundsen and Bellingshausen Seas. In contrast, these positive SIE errors have been greatly reduced in the ANTSIC-UNet predictions. ANTSIC-UNet outperforms the linear trend model throughout the year for all the lead times and most regions, 250 except for the Amundsen and Bellingshausen Seas. This is also true for 2017 and 2022 (Figs. S3 and S4). Therefore, ANTSIC-UNet has good predictive skills for extreme events in recent years.

|  |  | ANTSIC-Unet | Linear trend | Anomaly persistence |
|---|---|---|---|---|
| 2017 | RMSE | 0.21 | 0.25 | 0.24 |
|  | IIEE | 1.80 | 2.56 | 2.52 |
| 2022 | RMSE | 0.21 | 0.22 | 0.23 |
|  | IIEE | 1.68 | 2.24 | 2.45 |
| 2023 | RMSE | 0.24 | 0.27 | 0.31 |
|  | IIEE | 2.00 | 3.05 | 3.11 |

**Table 3. The averaged predictive skill of ANTSIC-Unet, linear trend and anomaly persistence models for the extremely low sea ice extent years of Antarctic sea ice.**






**Figure 7. February and September SIC of NSIDC observations (a, e) and errors predicted by the linear trend model (c, g), anomaly persistence model (d1-d3, h1-h3) and ANTSIC-UNet (b1-b3, f1-f3) at lead time of 1, 3 and 5 months for 2023 (lowest sea ice extent on record).**





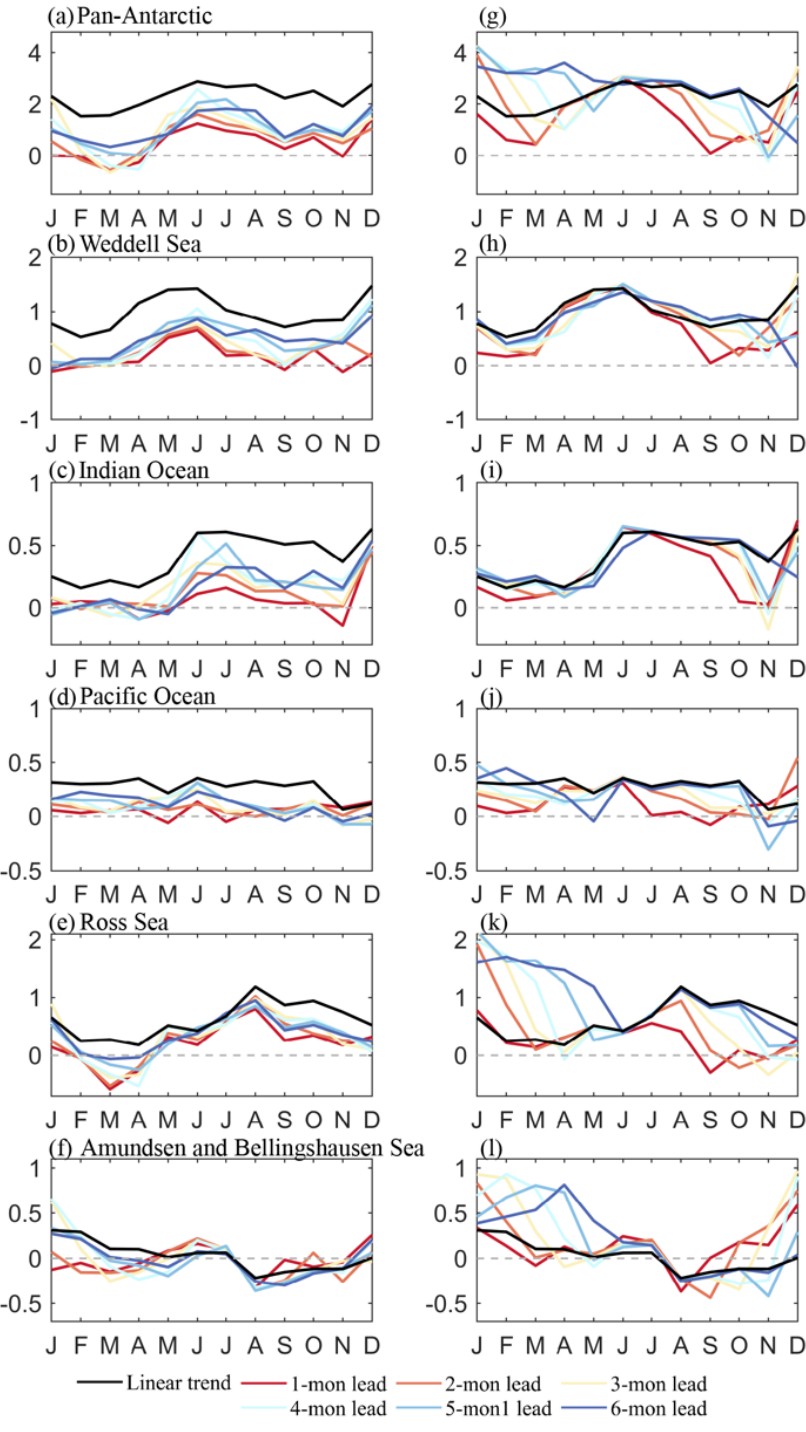


**Figure 8. Seasonality errors of the Pan- and regional Antarctic monthly mean SIE (SIC > 15%) between NSIDC observations and ANTSIC-UNet (a-f) and anomaly persistence model (g-l) predictions at different lead times for 2023 (lowest sea ice extent on record). The black lines show the seasonality SIE errors between observations and linear trend model. (units: million square kilometers)**



### 3.4 Variable importance

Previous studies suggested that the evaluation metrics of the predictive skill of a particular model with excellent generalization ability are strongly correlated to feature importance (FI) (Andersson et al., 2021; Molnar, 2019). The permutation feature importance method based on testing data can help us to figure out the model-dependence variables and explain the contribution extent of the variables to the performance of the model on unseen data. As discussed above, ANTSIC-UNet shows better performance compared to the linear trend and anomaly persistence models. This implies that ANTSIC-UNet has learned to

predict extended seasonal Antarctic sea ice based on the physical relationships of the input variables. Here we use the permutation feature importance measurement to explain model variance based on the testing data from 2020-2023. The variable importance results of the Pan-Antarctic averaged for all calendar months (Fig. 9) indicate that ANTSIC-UNet has learned to use some important variables, including sea ice conditions, sea surface temperature, radiative flux, and stratospheric wind. ANTSIC-UNet has also learned to ignore some peripheral variables, such as sea level pressure and subsurface ocean

temperature. At short lead times, on timescales of up to two months, ANTSIC-UNet relies more on the initial sea ice state and linear trend prediction, as well as the surface upward shortwave radiation, sea surface temperature, atmospheric conditions in the troposphere, and 10-hPa zonal wind in the stratosphere. This implies that ANTSIC-UNet has learned the dynamic and thermodynamic physical mechanisms directly forcing sea ice variations (Son et al., 2009; Turner et al., 2016). At longer lead times, in addition to historical SIC conditions and linear trend predictions of SIC at the target month, the 10-hPa zonal wind

stands out as an important influencing factor which manifests the lagged response in Antarctic sea ice to changes in stratospheric circulation. (Raphael and Hobbs, 2014; Wang et al., 2021).







**Figure 9. The results of variable importance analysis for Pan-Antarctic based on the permutation feature importance measurement (see Table 1 for full name of the variables).**



## 4 Discussion and Conclusion

In this study, we have introduced a deep learning model, ANTSIC-UNet, to predict the extended seasonal Antarctic monthly-mean sea ice concentration. Considering the complex physical processes influencing Antarctic sea ice variability, not only sea ice itself but also related atmospheric and oceanic variables are used for ANTSIC-UNet's forecasts. We compare the deep learning predictions against two benchmark models, the linear trend and anomaly persistence models, to evaluate the predictive skill of both Pan- and regional Antarctic sea ice. ANTSIC-UNet exhibits superior predictive skill for Antarctic sea ice at least 6 months lead and particularly improved predictions of extreme low sea ice extent events in recent years. The prediction performance of ANTSIC-UNet shows pronounced seasonality and regional dependence, which leads to the limitation of the predictive skill of the Pan-Antarctic. Specifically, during the autumn to spring, low RMSE are observed for most sub-regions. However, increased RMSE is evident in summer as the lead time exceeds 2 months indicating the decreased model performance. Small values of integrated ice-edge error (IIEE) are found in summer at 1-3 month lead, but large errors occur from November to January as the lead time exceeds 2-4 months. The low RMSE and broader coverage of small IIEE suggest superior predictive skills in the Pacific and Indian Oceans at the sea ice edge zone in summer. We further assess the prediction performance for year-to-year variability, ANTSIC-UNet shows good predictive skill in capturing the interannual variability of Pan-Antarctic and regional sea ice extent anomalies. Consistently high values of ACC and MSSS seen in the Weddell and Ross Seas suggest the possibility of long-lead extended seasonal predictions. Moreover, the results from the variable importance analysis, computed by a post-hoc interpretation method, suggest that ANTSIC-UNet has learned the relationship between the sea ice and other model-dependence climate variables with varying impacts across different lead times. Specifically, at short lead times, ANTSIC-UNet predictions are sensitive to initial conditions and linear trend predictions of SIC, sea surface temperature, radiative flux and vertical atmospheric circulation conditions. At longer lead times, predictions are dependent on historical conditions and linear trend predictions of SIC, and stratospheric circulation patterns. Amundsen and Bellingshausen Seas have the lowest predictive skill might be associated with that ANTSIC-UNet ignores the sea level pressure and does not consider the relationships of the tropical teleconnection and strengthening of Amundsen Sea Low (ASL) in recent decades (Li et al., 2021; Cai et al., 2023).

In addition, the ANTSIC-UNet model is trained on minimizing the loss function which measures the difference between the output and the desired targets. We optimize ANTSIC-UNet using the mean square error (MSE) of SIC as its original loss function. However, the pronounced prediction errors often occur at the sea ice edge, which might be associated with oceanic influence and wind dynamics. Y. Ren and X. Li (2023) suggested that the normalized integrated ice-edge error loss might be suitable for long sequence SIC predictions. The question is whether physically constrained loss function in deep learning models can improve the extended seasonal forecast of Antarctic sea ice. Here we test a hybrid loss function combining MSE and IIEE to optimize spatial predictions and minimize sea ice edge errors. IIEE loss is calculated by dividing the difference between the predicted and observed sea ice extent by the sum of SIE where SIC > 0.15% in both the prediction and observation.



We assign a weight of 0.05 to the IIEE components for values balance in the hybrid loss. The two loss functions are calculated as:

$$Original\ Loss = MSE = mean(\textstyle\sum(p - o)^2), \tag{9}$$

$$Hybrid\ Loss = MSE + 0.05\frac{IIEE}{SIE_p \cup SIE_o}, \tag{10}$$

where $p$ ($SIE_p$) is thr predicted sea ice concentration (ice extent) by ANTSIC-UNet and $o$ ($SIE_o$) is the observed ice concentration (ice extent). For clarity, we denote the original loss (hybrid loss) as subscripts "o" ("h") for distinguish between the ANTSIC-UNet models trained with two different loss functions.

Our results show similar distributions of sea ice edge errors predicted by two ANTSIC-UNet models (Fig. 4 a2-f2 and Fig. 10
a1-f1) with small values of IIEE at 1-month lead and large values from November to January as the lead time exceeds 2-4 months. ANTSIC-UNet_h trained with the hybrid loss slightly reduces the IIEE for the Pan-Antarctic compared to ANTSIC-UNet_o, especially in Weddell Ocean, Ross Amundsen and Bellingshausen Seas (~0.02-0.05 million km$^2$). However increased errors occur in these regions as lead time exceeds 3-4 months (Fig. 10 a2-f2).

Thus ANTSIC-UNet provides a useful tool for extended seasonal prediction of Antarctic sea ice concentration and extent,
which also provides valuable information for analyzing the influencing factors of sea ice variations in different regions. Further investigation is needed based on physically enriched deep learning models, i.e., a better understanding of physical mechanisms between SIC and other climate variables with long-term memory such as sea ice thickness and ocean heat content (Marchi et al., 2019; Bushuk et al., 2021; Libera et al., 2022).



**Figure 10. The IIEE of ANTSIC-UNet_h (a1-f1) and difference (b2-f2) between the two ANTSIC-UNet models trained with different loss functions for different target months and forecast lead times spatially and temporally averaged during the testing years. (units: million square kilometers)**



*Data Availability.* All the data analyzed here are openly available. NSIDC sea ice concentration data is publicly available at https://nsidc.org/data/nsidc-0079/versions/3. ERA5 monthly averaged data on pressure levels from 1979 to present is publicly available at https://cds.climate.copernicus.eu/doi/10.24381/cds.6860a573. ERA5 monthly averaged data on single levels from 1979 to present is publicly available at https://cds.climate.copernicus.eu/doi/10.24381/cds.f17050d7. ORAS5 monthly average data from 1979 to present is publicly available at https://cds.climate.copernicus.eu/cdsapp#!/dataset/10.24381/cds.67e8eeb7.

*Author contributions.* JL conceived the study, ZY and JL designed the model, carried out the analysis and wrote the paper; all authors participated in constructive discussions and helped improve the manuscript.

*Competing interests.* The authors declare that they have no conflict of interest.

*Acknowledgements.* This research is supported by National Key Research and Development Program of China(Grant No. 2022YFE0106800).

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
