# Peer review of "Extended seasonal prediction of Antarctic sea ice concentration using ANTSIC-UNet"

_EGUsphere, 2024_

## Author Comment (AC1)

**Response to comments by Reviewer #1**

We would like to thank the reviewer for the helpful comments on the manuscript. Please find below our responses to the comments.

*This article introduces a deep learning model called ANTSIC-UNet for predicting the extended seasonal variations in Antarctic sea ice concentration, with the ability to forecast up to 6 months in advance. The study utilized a rich set of climate variables for model training and compared it against two benchmark models (linear trend and anomaly persistence models). The results demonstrate that ANTSIC-UNet exhibits superior predictive skills in sea ice concentration and integrated ice-edge error, especially in forecasting extreme events in recent years. The strengths of the article include the consideration of both sea ice and related atmospheric and oceanic variables enhances the accuracy of the predictions. The results are interesting and the work could be published after moderate revision. My comments are intended to improve the presentation of the paper and require clarifying unclear points.*

*Comments*
*1. L76 "57 is the dimension of the variables" However, when we calculate 12+1+14\*3+1, it equals 56. So, what is the extra one?*

The dimension of 57 includes sea ice concentration for the past 12 months, the linear trend prediction of sea ice concentration for the following 6 months, 12 climate variables for the past 3 months, 2 climate variables for the past 1 month, and the land mask. Therefore, the calculation is $12 + 6 + 12*6 + 2 + 1 = 57$. For the details of climate variables, please refer to Table 1, which provides the variable names along with their respective lead or lag times. We clarified this in the revision.

*2. L184 For September, compared to anomaly persistence, ANTSIC-UNet shows a larger negative bias in the sea ice edge region. What could be the possible reasons for this error?*

Thanks for your comment. The larger negative bias in the sea ice edge region in September for the ANTSIC-UNet prediction relative to the anomaly persistence as the lead time increase is due to the limited number of years used for calculating the average of sea ice errors, which only includes the testing years of 2017, 2020 to 2023 (anomalously low ice extents), and 2014 (record high). Specifically, this averaging results in large positive and negative anomalies in different years offsetting each other for the anomaly persistence prediction. To demonstatrate this, we selected three sub-regions that show larger negative bias in the sea ice edge region in September for ANTSIC-UNet at 5-month lead compared to the anomaly persistence prediction (see Figure R1), including the Weddell Sea, the Pacific Ocean, and the Amundsen and Bellingshausen Seas. Here we used the mean absolute error (MAE) as the evaluation metrics (Figure R2). ANTSIC-UNet shows smaller prediction errors in the sea ice edge across all regions compared to anomaly persistence, except for the Weddell Sea as the lead time exceeds 4 months.

[Figure]

Figure R1. September mean sea ice concentration errors predicted by (a) ANTSIC-UNet and (b) anomaly persistence model at 5-month lead for the testing years. The red boxes indicate the three regions where ANTSIC-UNet shows larger negative bias compared to the anomaly persistence model: region 1 – eastern Weddell Sea (53°-63°S, 20°W-30°E), region 2 – eastern Pacific Ocean (60°-65°S, 115°-160°E) and region 3 - Amundsen and Bellingshausen Seas (62°-72°S, 130°-60°W).

[Figure]

Figure R2. September sea ice concentration mean absolute error (SIC MAE) between the predictions and NSIDC observations for (a) eastern Weddell Sea, (b) eastern Pacific Ocean, and (c) Amundsen and Bellingshausen Seas for the testing years. (ANTSIC-UNet: red line; anomaly persistence model: blue line)

3. *L186 Is the lower RMSE in September compared to February related to the size of the area considered during the calculation? Are the regions used for calculating each indicator consistent with the respective months?*

Thanks for your comment. The RMSE is calculated based on the area where sea ice concentration is more than 15% in observations or predictions. The IIEE is the sum of overestimated and underestimated sea ice extent where sea ice concentration is more than 15%.

The area size varies in different months. Both RMSE and IIEE with the respective months are measured in units of area. Figure R3 shows the percentage of the sea ice edge error relative to

the actual sea ice extent. In February, although the Antarctic sea ice extent reaches its seasonal minimum, the relative percentage of the sea ice edge error is large and increases as lead times increase. In September, the Antarctic experiences extensive sea ice coverage, but the relative percentage is smaller for all lead months, resulting in the overall low RMSE.

[Figure]

Figure R3. Percentage of the sea ice edge error relative to the actual sea ice extent (where sea ice concentration more than 15% in observations or predictions).

4. *L274 Is the high importance of variables in the* model due to the seasonal cycle? Does the importance of variables change for SIC anomaly?

Thanks for your comment. The importance of relevant climate variables is independent of the seasonal cycle. All non-SIC variables, were converted to anomalies (by subtracting the climatological mean for each calendar month during 1979-2011) before being input into ANTSIC-UNet. SIC has a pronounced seasonal cycle, which serves as an important reference for predicting future changes.

Variable importance changes across different seasons in the context of SIC anomaly. For example, Table R1 (see below) gives the variable importance ranking for the target months of January and June at 1-month lead. For January, ANTSIC-UNet relies mostly on the upwelling solar radiation and 10-hPa zonal wind in the stratosphere. For June, sea surface temperature and initial sea ice state are more important. Additionally, the linear trend predictions of SIC at the target month are important for both months though it ranks as the third.

Table R1. Variable importance ranking for the target months of January and June at 1-month lead averaged for the testing years 2020-2023.

| Rank | (a) For Jan forecasts | (b) For Jun forecasts |
|------|------------------------|------------------------|
| 1 | Dec USRA (0.90%) | May SSTA (0.79%) |
| 2 | Dec U10hPaA (0.55%) | May SIC (0.46%) |
| 3 | Jan SIC trend(0.46%) | Jun SIC trend (0.35%) |

5. *The main improvement of this article compared to other DL methods is the inclusion of relevant variables that affect sea ice in the training data of the model. How significant is the impact of these variables compared to a model trained solely using historical data?*

Thanks for this question and concern about the improvement made by incorporating relevant variables for training. We did compare the performance of DL models trained by three sets of input variables (Table R2).

As shown in Table R2, compared to HIS-V which is trained by historical data without incorporating the future 6 months linear trend predictions of sea ice concentration, ANTSIC-UNet shows relatively reduced RMSE and notable improvement in IIEE during all testing years and extreme years. This suggests that incorporating future sea ice trends enhances the deep-learning model's predictive accuracy, particularly at the sea ice edge.

Furthermore, compared to SIC-V, which is trained by only sea ice data, including the future 6 months linear trend predictions and past 12 months of historical sea ice concentration, Both ANTSIC-UNet and HIS-V show significant improvement of IIEE, which indicates that using enriched climate variables as inputs allows ANTSIC-UNet to effectively capture the complex nonlinear relationships in air-ice-sea interactions and enhance the predictive skill for Antarctic sea ice concentration.

Table R2. The averaged predictive skill of ANTSIC-UNet (the original DL model trained by 57 variables, see Table 1 in the manuscript for the details of all input variables), HIS-V (DL model trained by historical data, without incorporating the future 6 months linear trend predictions of sea ice concentration), and SIC-V (DL model trained by pure SIC data, including the future 6 months linear trend predictions of sea ice concentration and past 12 months of historical sea ice concentration). (RMSE: root-mean-square error; IIEE: integrated ice-edge error.)

| | | ANTSIC-UNet | HIS-V | SIC-V |
|---|---|---|---|---|
| All testing years | RMSE | 0.21 | 0.22 | 0.22 |
| | IIEE | 1.68 | 1.75 | 1.95 |
| 2017 | RMSE | 0.21 | 0.22 | 0.22 |
| | IIEE | 1.80 | 1.92 | 2.27 |
| 2022 | RMSE | 0.21 | 0.22 | 0.22 |
| | IIEE | 1.68 | 1.77 | 1.98 |
| 2023 | RMSE | 0.24 | 0.25 | 0.24 |
| | IIEE | 1.99 | 2.07 | 2.57 |

6. *The section on the importance of each variable is very insightful. The author presents some viewpoints that are inconsistent with statistical models, such as the minimal impact of variables like temperature and wind speed in DL methods. Does this suggest that DL methods have not learned the underlying mechanisms of these variables to some extent?*

Thank you for your comment. Our study showed that ANTSIC-UNet had been trained to learn the nonlinear and indirect relationships among climate variables that contribute to improved

accuracy of Antarctic sea ice prediction. The variable importance results from ANTSIC-UNet are generally consistent with known causal links between climate variables and sea ice, suggesting that physically plausible statistical relationships have been learned. For example, sea surface temperature and air temperature play a crucial role in Antarctic sea ice predictions at 1-2 month lead, influencing sea ice formation and melting through thermodynamic processes. 10m meridional wind is also important at short lead times, affecting sea ice variation through sea ice advection, air-sea heat flux, and ocean mixing. As the lead time increases, the influence of these variables tends to be reduced, and the 10-hPa zonal wind in the stratosphere becomes more important. This is consistent with previous studies showing that the changes in stratospheric zonal circulation affect sea ice variability by influencing the circumpolar westerly winds in the troposphere through downward propagation (Wang et al., 2019; Cordero et al., 2023). The relatively not very significant importance of tropospheric variables (i.e., H500A) may be related to the inherent structure of the deep learning model that still has not learned all underlying mechanisms, which requires further investigation in future research.

When a variable shows small or even negative importance, as Andersson et al. (2021) suggested the DL model might be overlooking that feature or has not yet fully captured the intrinsic relationships involving that variable.

Reference:
Andersson, T. R., Hosking, J. S., Pérez-Ortiz, M., Paige, B., Elliott, A., Russell, C., Law, S., Jones, D. C., Wilkinson, J., Phillips, T., Byrne, J., Tietsche, S., Sarojini, B. B., Blanchard-Wrigglesworth, E., Aksenov, Y., Downie, R., and Shuckburgh, E.: Seasonal Arctic sea ice forecasting with probabilistic deep learning, Nature Communications, 12, 5124, https://doi.org/10.1038/s41467-021-25257-4, 2021.

Cordero, R. R., Feron, S., Damiani, A., Llanillo, P. J., Carrasco, J., Khan, A. L., Bintanja, R., Ouyang, Z., and Casassa, G.: Signature of the stratosphere–troposphere coupling on recent record-breaking Antarctic sea-ice anomalies, The Cryosphere, 17, 4995–5006, https://doi.org/10.5194/tc-17-4995-2023, 2023.

Wang, G., Hendon, H. H., Arblaster, J. M., Lim, E.-P., Abhik, S., and van Rensch, P.: Compounding tropical and stratospheric forcing of the record low Antarctic sea-ice in 2016, Nat Commun, 10, 13, https://doi.org/10.1038/s41467-018-07689-7, 2019.

---

## Author Comment (AC4)

**Response to comments by Reviewer #2**

We would like to thank the reviewer for the helpful comments on the paper. Please find below our responses to the comments.

*This paper documents the results from a deep learning effort at predicting maps of Antarctic sea ice from the NSIDC. The model is generally well-described, with well documented results that are effectively compared to simple linear trends and anomaly persistence. However, the paper focus is only on the performance of a single effort of sea ice prediction and contains no effort to use this tool to add any scientific knowledge or insight to Cryospheric science. There is only the briefest attempt to contextualise the importance of Antarctic Sea Ice prediction, and the reasoning behind the variable selection is not described at all. There is very little documentation on how the model was developed and any insight into what was learnt during the development process. The publishing criteria for the Cryosphere is that there needs to be a scientific aspect to the publications beyond model description and results, therefore this paper is not acceptable=and in opinion needs to be rejected.*

*Particular issues:*
*Throughout the paper there is a lack of knowledge of the system that is being investigated, and the study is only focused or representing the input data and the physical system. For example the title says it 'predicts' sea ice – there are many aspects of sea ice that are not considered here. This paper only looks at monthly sea ice concentration maps from the NSIDC – possibly the simplest representation of sea ice. There are many other datasets available – this needs to be documented. The introduction is very brief and contains no description of the system being investigated.*

Thank you for your comments. Firstly, we modified the introduction to emphasize the importance of accurate predictions for Antarctic sea ice concentration. Compared to the Arctic, the prediction of Antarctic sea ice has received much less attention. Also, the demand for subseasonal to extended seasonal Antarctic sea ice predictions has been recognized due to the expanding range of activities in the Southern Ocean (Zampieri et al., 2019; Bushuk et al., 2021; Libera et al., 2022). Accurate sea ice concentration predictions can provide early warnings about sea ice changes and related hazards. This is particularly important for managing the risks of shipping activities in the Southern Ocean. For example, two polar vessels, Akademik Shokalskiy and Xuelong became trapped in rapidly formed sea ice in the Antarctic coastal region (Wang et al., 2014). Commercial fishing and tourism operations mostly use ice-strengthened vessels rather than icebreakers, which are vulnerable to sea ice hazards. It also supports ecosystem management and informs policy decisions, since the seasonal variations in Antarctic sea ice have a profound influence on marine productivity and fisheries (Libera et al., 2022).

Secondly, the deep learning model developed for Antarctic sea ice concentration predictions has been described in Figure R1. To further address your concerns, we have included more details about the model system being developed. A U-shaped architecture based on convolutional neural networks is widely used for many applications, i.e., remote sensing image

segmentation tasks (Marmanis et al., 2016; Wang et al., 2023). Recently, Andersson et al. (2021) employed the U-Net for three-class predictions of Arctic sea ice concentration. For accurate forecasts of Antarctic sea ice concentration, we made necessary modifications to the original architecture of U-Net and turned it into single value regression rather than the classification. The ANTSIC-UNet's inputs are feature maps of high-resolution sea ice concentration, other multiple climate variables related to sea ice changes over different lead/lag months and a land mask. The outputs are high-resolution sea ice concentration maps for the future months. The inputs are processed into a large number of feature maps with decreased dimensionality by the encoder part of ANTSIC-UNet. Such deep layers and large-scale features allow the model to capture complex nonlinear relationships and provide an interpretation of the inputs. The decoder then upscales the feature maps extracted by the encoder into upsampled features and uses four skip connections to combine them with multi-scale features from different scale levels of the encoder. This process results in high-resolution output maps that align with the spatial dimensions of the input data. Sigmoid activation functions are used in the final six convolutional layers to generate regression predictions of Antarctic sea ice concentration maps for six months. There are also other attempts in the training algorithm for enhancing the predictive skill of the proposed model, for example, the hybrid loss function combining sea ice concentration mean square error (MSE) and integrated ice-edge error (IIEE) (see details in Section 4). The results presented evidence that models trained with this approach predict more accurately at the sea ice edge, thereby improving prediction performance.

[Figure]

Figure R1. Configuration of ANTSIC-UNet model used for extended seasonal Antarctic sea ice prediction. Inputs are sea ice concentration, other climate variables related to sea ice

changes over different lead/lag months and a land mask. The U-shaped architecture includes the encoder, decoder and four skip connections. Sigmoid activation functions (*fs*) are used in the final six convolutional layers to generate regression predictions of Antarctic sea ice concentration maps for six months.

Thirdly, this work is motivated by the fact that the Antarctic sea ice extent exhibits significant variability driven by the complex air-ice-sea interactions that are not yet fully understood. In this study, we clarified how each climate variable contributes to sea ice variation selected for the training of ANTSIC-UNet and explored which specific variable plays more important roles in the different months of sea ice prediction. Our results show evidence that ANTSIC-UNet can successfully extract key information from the complex ocean-ice-atmosphere interactions to predict sea ice concentration and capture seasonal variations through the different important climate variables. This approach could be effectively extended to other sea ice variables once the relevant long-term data becomes available (i.e., sea ice thickness). This potential for broader applicability underscores the significance of our work and its contribution to advancing Antarctic sea ice predictions.

Finally, sea ice concentration is the essential variable for investigating the variation of sea ice (i.e., extent) and the satellite observation provides long-term records of the data. Thus, our study focused on sea ice concentration, and used monthly maps from the NSIDC which provides long-term records of data for the training of deep learning models since the late 1970s. Following the reviewer's suggestion, in the discussion, we further documented other available datasets and discussed the potential for extending our research by integrating these additional datasets into future studies. These include Antarctic sea ice thickness data from satellite altimetry missions including the ICESat data (from 2003-2008), ICESat-2 data (from late 2018 onward) and CryoSat-2 data (from 2010 onward) which remains limited in terms of confidence and temporal coverage which are not yet suitable for deep learning applications (Hendricks et al., 2018; Kacimi and Kwok, 2020; Fons et al., 2023). The Polar Pathfinder product (Tschudi et al. 2019) provides daily sea ice motion vectors at a spatial resolution of 25 km which is valuable for investigating sea ice movement patterns under the influence of wind and ocean currents. In future research, we will explore whether incorporating ice drift can enhance the accuracy of sea ice predictions.

Reference:

Andersson, T. R., Hosking, J. S., Pérez-Ortiz, M., Paige, B., Elliott, A., Russell, C., Law, S., Jones, D. C., Wilkinson, J., Phillips, T., Byrne, J., Tietsche, S., Sarojini, B. B., Blanchard-Wrigglesworth, E., Aksenov, Y., Downie, R., and Shuckburgh, E.: Seasonal Arctic sea ice forecasting with probabilistic deep learning, Nature Communications, 12, 5124, https://doi.org/10.1038/s41467-021-25257-4, 2021.

Bushuk, M., Winton, M., Haumann, F. A., Delworth, T., Lu, F., Zhang, Y., Jia, L., Zhang, L., Cooke, W., Harrison, M., Hurlin, B., Johnson, N. C., Kapnick, S. B., McHugh, C., Murakami, H., Rosati, A., Tseng, K.-C., Wittenberg, A. T., Yang, X., and Zeng, F.: Seasonal Prediction and Predictability of Regional Antarctic Sea Ice, Journal of Climate, 34, 6207–6233, https://doi.org/10.1175/JCLI-D-20-0965.1, 2021.

Libera, S., Hobbs, W., Klocker, A., Meyer, A., and Matear, R.: Ocean-Sea Ice Processes and Their Role in Multi-Month Predictability of Antarctic Sea Ice, Geophysical Research Letters, 49, e2021GL097047, https://doi.org/10.1029/2021GL097047, 2022.

Marmanis, D., Datcu, M., Esch, T., and Stilla, U.: Deep Learning Earth Observation Classification Using ImageNet Pretrained Networks, IEEE Geoscience and Remote Sensing Letters, 13, 105–109, https://doi.org/10.1109/LGRS.2015.2499239, 2016.

Fons, S., Kurtz, N., and Bagnardi, M.: A decade-plus of Antarctic sea ice thickness and volume estimates from CryoSat-2 using a physical model and waveform fitting, The Cryosphere, 17, 2487–2508, https://doi.org/10.5194/tc-17-2487-2023, 2023.

Hendricks, S., Paul, S., and Rinne, E.: ESA Sea Ice Climate Change Initiative (Sea_Ice_cci): Southern hemisphere sea ice thickness from CryoSat-2 on the satellite swath (L2P), v2.0, Centre for Environmental Data Analysis [data set], https://doi.org/10.5285/fbfae06e787b4fefb4b03cba2fd04bc3, 2018.

Kacimi, S. and Kwok, R.: The Antarctic sea ice cover from ICESat-2 and CryoSat-2: freeboard, snow depth, and ice thickness, The Cryosphere, 14, 4453–4474, https://doi.org/10.5194/tc-14-4453-2020, 2020.

Tschudi, M., Meier, W. N., Stewart, J. S., Fowler, C., and Maslanik, J.: Polar Pathfinder Daily 25 km EASE-Grid Sea Ice Motion Vectors, Version 4, Boulder, CA, USA, NASA National Snow and Ice Data Center Distributed Active Archive Center, https://doi.org/10.5067/INAWUWO7QH7B, 2019

Wang, X., Hu, Z., Shi, S., Hou, M., Xu, L., and Zhang, X.: A deep learning method for optimizing semantic segmentation accuracy of remote sensing images based on improved UNet, Sci Rep, 13, 7600, https://doi.org/10.1038/s41598-023-34379-2, 2023.

Wang, Z., Turner, J., Sun, B., Li, B., and Liu, C.: Cyclone-induced rapid creation of extreme Antarctic sea ice conditions, Sci Rep, 4, 5317, https://doi.org/10.1038/srep05317, 2014.

Zampieri, L., Goessling, H. F., and Jung, T.: Predictability of Antarctic Sea Ice Edge on Subseasonal Time Scales, Geophysical Research Letters, 46, 9719–9727, https://doi.org/10.1029/2019GL084096, 2019.

*The most useful aspect of the study can be to inform of what variables from the chosen reanalysis are the strongest predictors. This is attempted in section 3.4 – but it has no contextualization. Key aspects that need including: Why physically may each variable be useful in prediction? How accurate are each variable within the reanalysis product? The lack of predictive importance for Downward solar for example may be due to this variable being poorly represented within the reanalysis. What other scientific analysis has been performed using this reanalysis? How has it been used outside of Deep learning to investigate sea ice?*

Thank you for your comments. In this revision, we have made several improvements to address your concerns.

Firstly, we provided an overview of how reanalysis products have been applied to sea ice investigations outside of deep learning and summarized the representation accuracy within the chosen reanalysis products. Reanalysis products are vital tools for studying climate variability in the Antarctic due to the sparse observations. They are widely used as inputs of dynamical models, serving as initial and boundary conditions, and are also crucial for validating model simulations and predictions (Hobbs et al., 2020; Goosse et al., 2023; Mezzina et al., 2024). The ECWMF Reanalysis v5 (ERA5) data is generally considered the best-performing atmospheric reanalysis dataset for polar regions. Previous studies have extensively evaluated the performance of ERA5, which accurately represents near-surface wind, temperature, and sea level pressure in the Antarctic (Gossart et al., 2019; Tetzner et al., 2019; Andres-Martin et al., 2024). However, deficiencies in cloud cover and water content have resulted in significant surface radiation biases during the austral summer, particularly due to the underestimation of cloud cover (Wang et al., 2020; Mallet et al., 2023). The limited observational data in the mid-to-upper troposphere and the stratospheric leads to certain uncertainty in mid- and high-level pressure and temperature, and the representation of the stratospheric polar vortex (Orr et al., 2021). In addition to atmospheric reanalysis, oceanic reanalysis products like Ocean Reanalysis System 5 (ORAS5) are crucial for understanding the principal mechanism of the Southern Ocean. ORAS5 has been shown to effectively capture sea surface temperatures in the Antarctic, with the vertical temperature structure also aligning closely with observations (Cai et al., 2023).

Secondly, we elaborated on the physical relevance of each variable for predicting sea ice concentration. In our study, 14 atmospheric and oceanic variables from ERA5 and ORAS5 are selected to capture the key physical mechanisms influencing sea ice variations. Variables such as sea surface temperature, 2m air temperature, and radiation impact heat flux exchanges at the air-ice-sea interface (Bourassa et al., 2013). Near surface winds drive sea ice movement and large-scale tropospheric circulation impacts sea ice through its effects on winds, temperature, precipitation, and cloud cover (Raphael and Hobbs, 2014). The 10-hPa zonal wind represents stratospheric zonal circulation, which impacts surface circulation through downward propagation, influencing sea ice dynamics (Cordero et al., 2023). Sea temperature anomalies and the upper-ocean heat content anomaly for the upper 300 m taken from ORAS5 play a crucial role in the heat energy exchange at the ocean–ice interface (Purich and Doddridge, 2023; Bianco et al., 2024). The upwelling of warmer subsurface water can further influence sea ice formation and melting in the high latitude of the Southern Ocean (Cai et al., 2023).

Finally, we discussed the reasons for the lack of predictive importance of variables such as downward solar radiation in ANTSIC-UNet. When a variable shows minimal or even negative importance, it suggests that the ANTSIC-UNet might be overlooking that feature or has not yet fully captured the intrinsic relationships involving that variable. It may also be related to the accuracy of the reanalysis data used as input. For example, the lack of predictive importance for downward solar radiation could be due to this variable being poorly represented in the Southern Ocean within the reanalysis as discussed above. Thus, it is crucial to consider the accuracy of input variables chosen from reanalysis data for Antarctic sea ice predictions.

Reference:

Andres-Martin, M., Azorin-Molina, C., Serrano, E., González-Herrero, S., Guijarro, J. A., Bedoya-Valestt, S., Utrabo-Carazo, E., and Vicente Serrano, S. M.: Near-surface wind speed trends and variability over the Antarctic Peninsula, 1979–2022, Atmospheric Research, 309, 107568, https://doi.org/10.1016/j.atmosres.2024.107568, 2024.

Bianco, E., Iovino, D., Masina, S., Materia, S., and Ruggieri, P.: The role of upper-ocean heat content in the regional variability of Arctic sea ice at sub-seasonal timescales, The Cryosphere, 18, 2357–2379, https://doi.org/10.5194/tc-18-2357-2024, 2024.

Bourassa, M. A., Gille, S. T., Bitz, C., Carlson, D., Cerovecki, I., Clayson, C. A., Cronin, M. F., Drennan, W. M., Fairall, C. W., Hoffman, R. N., Magnusdottir, G., Pinker, R. T., Renfrew, I. A., Serreze, M., Speer, K., Talley, L. D., and Wick, G. A.: High-Latitude Ocean and Sea Ice Surface Fluxes: Challenges for Climate Research, https://doi.org/10.1175/BAMS-D-11-00244.1, 2013.

Cai, W., Jia, F., Li, S., Purich, A., Wang, G., Wu, L., Gan, B., Santoso, A., Geng, T., Ng, B., Yang, Y., Ferreira, D., Meehl, G. A., and McPhaden, M. J.: Antarctic shelf ocean warming and sea ice melt affected by projected El Niño changes, Nat. Clim. Chang., 13, 235–239, https://doi.org/10.1038/s41558-023-01610-x, 2023.

Cordero, R. R., Feron, S., Damiani, A., Llanillo, P. J., Carrasco, J., Khan, A. L., Bintanja, R., Ouyang, Z., and Casassa, G.: Signature of the stratosphere–troposphere coupling on recent record-breaking Antarctic sea-ice anomalies, The Cryosphere, 17, 4995–5006, https://doi.org/10.5194/tc-17-4995-2023, 2023.

Goosse, H., Allende Contador, S., Bitz, C. M., Blanchard-Wrigglesworth, E., Eayrs, C., Fichefet, T., Himmich, K., Huot, P.-V., Klein, F., Marchi, S., Massonnet, F., Mezzina, B., Pelletier, C., Roach, L., Vancoppenolle, M., and van Lipzig, N. P. M.: Modulation of the seasonal cycle of the Antarctic sea ice extent by sea ice processes and feedbacks with the ocean and the atmosphere, The Cryosphere, 17, 407–425, https://doi.org/10.5194/tc-17-407-2023, 2023.

Gossart, A., Helsen, S., Lenaerts, J. T. M., Broucke, S. V., Lipzig, N. P. M. van, and Souverijns, N.: An Evaluation of Surface Climatology in State-of-the-Art Reanalyses over the Antarctic Ice Sheet, https://doi.org/10.1175/JCLI-D-19-0030.1, 2019.

Hobbs, W. R., Klekociuk, A. R., and Pan, Y.: Validation of reanalysis Southern Ocean atmosphere trends using sea ice data, Atmospheric Chemistry and Physics, 20, 14757–14768, https://doi.org/10.5194/acp-20-14757-2020, 2020.

Mallet, M. D., Alexander, S. P., Protat, A., and Fiddes, S. L.: Reducing Southern Ocean Shortwave Radiation Errors in the ERA5 Reanalysis with Machine Learning and 25 Years of Surface Observations, https://doi.org/10.1175/AIES-D-22-0044.1, 2023.

Mezzina, B., Goosse, H., Klein, F., Barthélemy, A., and Massonnet, F.: Atmospheric drivers of Antarctic sea ice extent summer minima, The Cryosphere Discussions, 1–20, https://doi.org/10.5194/tc-2023-45, 2023.

Orr, A., Lu, H., Martineau, P., Gerber, E. P., Marshall, G. J., and Bracegirdle, T. J.: Is our dynamical understanding of the circulation changes associated with the Antarctic ozone hole sensitive to the choice of reanalysis dataset?, Atmospheric Chemistry and Physics, 21, 7451–7472, https://doi.org/10.5194/acp-21-7451-2021, 2021.

Purich, A. and Doddridge, E. W.: Record low Antarctic sea ice coverage indicates a new sea ice state, Commun Earth Environ, 4, 1–9, https://doi.org/10.1038/s43247-023-00961-9, 2023.

Raphael, M. N. and Hobbs, W.: The influence of the large-scale atmospheric circulation on Antarctic sea ice during ice advance and retreat seasons, Geophysical Research Letters, 41, 5037–5045, https://doi.org/10.1002/2014GL060365, 2014.

Tetzner, D., Thomas, E., and Allen, C.: A Validation of ERA5 Reanalysis Data in the Southern Antarctic Peninsula—Ellsworth Land Region, and Its Implications for Ice Core Studies, Geosciences, 9, 289, https://doi.org/10.3390/geosciences9070289, 2019.

Wang, H., Klekociuk, A. R., French, W. J. R., Alexander, S. P., and Warner, T. A.: Measurements of Cloud Radiative Effect across the Southern Ocean (43° S–79° S, 63° E–158° W), Atmosphere, 11, 949, https://doi.org/10.3390/atmos11090949, 2020.

*Finally there is little to no contextualization of results amongst contemporary literature and other prediction efforts. Section 4 contains only a handful of citations when it is essential to contrast the results here with other efforts at sea ice predictions. How do the reported skills in forecasting compare to other efforts, Andersson et al. (2021) is an important bench mark here. How are the extreme years (2017, 2022, 2023) described in literature? What other hypothesis exist about what affected sea ice in these years?*

Thank you for your comments. Andersson et al. (2021) focused on Arctic sea ice prediction, comparing deep learning model performance at sea ice edge with the dynamic model and linear trend predictions, including extreme September sea ice events. Antarctic sea ice prediction has received less attention compared to the Arctic. To further assess the Antarctic sea ice predictive skill of ANTSIC-UNet against other prediction efforts, we included the dynamic model's monthly mean Antarctic sea ice concentration predictions calculated by the ensemble mean of 51 members of SEAS5, provided by the Copernicus Climate Change Service (C3S) Prediction project (Thépaut et al., 2018). SEAS5, ECMWF's fifth-generation seasonal forecast system, is recognized for its state-of-art predictive skill among the dynamical models which provides Antarctic sea ice concentration prediction for up to six months (Johnson et al., 2019). As shown in Figure R2, ANTSIC-UNet has small root-mean-square errors (RMSE) for Antarctic sea ice concentration, and outperforms the anomaly persistence predictions at all lead times. Compared to RMSE of SEAS5, ANTSIC-UNet shows slightly larger errors at 1-3 month lead, and smaller errors as lead time exceeds 4 months, which remains highly competitive. In terms of IIEE, ANTSIC-UNet shows significantly superior performance relative to all other models. The improvement in sea ice edge predictions of ANTSIC-UNet becomes more pronounced as the lead time increases.

[Figure]

Figure R2. The average predictive skill of Pan-Antarctic sea ice for ANTSIC-UNet, linear trend, anomaly persistence and SEAS5 predictions during the testing years. (a) SIC RMSE: root-mean-square error and (b) IIEE: integrated ice-edge error.

To our knowledge, little research has focused on the predictability of Antarctic sea ice extent in extreme years. We further compared the ANTSIC-UNet's accuracy performance on sea ice edge predictions for the extreme summer years, relative to linear trend predictions and SEAS5. As shown in Figure R3, both ANTSIC-UNet and SEAS5 have increasing sea ice edge errors as lead time increases. The linear trend predictions are independent of lead time. ANTSIC-UNet outperforms SEAS5 and linear trend predictions at sea ice edge error in all extreme summer years. At short lead times, ANTSIC-UNet has substantial improvement over the linear trend predictions and moderate improvement over SEAS5. At long lead times, ANTSIC-UNet's improvements over SEAS5 become more significant. These results suggest that ANTSIC-UNet has high predictive skills for extended seasonal predictions of Antarctic sea ice concentration, especially for extreme events, compared to other statistical and dynamic models.

[Figure]

Figure R3. Integrated ice-edge error (IIEE) of ANTSIC-UNet, the linear trend forecast and SEAS5 for February forecasts at lead time of 1, 3, and 5 months for the extreme summer years. (a) 2017, (b) 2022 and (c) 2023.

Antarctic sea ice has decreased in recent years, with summer sea ice coverage frequently reaching historic lows, including three extreme summer events. Some research have been carried out to investigate the key climate drivers and potential mechanisms behind these extreme conditions. The anomalous sea ice melting during the summer of 2017 might be associated with early spring atmospheric conditions over the Southern Ocean were primarily influenced by a positive phase of the zonal wave 3 (ZW3) pattern, followed by a near-record negative Southern Annular Mode (SAM) (Turner et al., 2017; Schlosser et al., 2018). The significant weakening of the polar stratospheric vortex was identified as a key driver of the SAM changes (Wang et al., 2019). The extremely low sea ice events in the summer of 2022 and 2023 occurred with the deepening of the Amundsen Sea Low (ASL), triggering feedbacks that played a crucial role in the reduction of summer sea ice (Turner et al., 2022; Wang et al., 2022). A few studies have emphasized that the influence of a warm subsurface ocean is a contributor to the recent record-low summer sea ice events (Liu et al., 2023; Purich and Doddridge, 2023). Different large-scale atmospheric circulation patterns may also lead to similar regional prevailing winds, driving the negative Antarctic sea ice extent anomalies (Mezzina et al., 2024).

Reference:

Mezzina, B., Goosse, H., Klein, F., Barthélemy, A., and Massonnet, F.: The role of atmospheric conditions in the Antarctic sea ice extent summer minima, The Cryosphere, 18, 3825–3839, https://doi.org/10.5194/tc-18-3825-2024, 2024.

Liu, J., Zhu, Z., and Chen, D.: Lowest Antarctic Sea Ice Record Broken for the Second Year in a Row, Ocean-Land-Atmosphere Research, 2, 0007, https://doi.org/10.34133/olar.0007, 2023.

Purich, A. and Doddridge, E. W.: Record low Antarctic sea ice coverage indicates a new sea ice state, Commun Earth Environ, 4, 1–9, https://doi.org/10.1038/s43247-023-00961-9, 2023.

Schlosser, E., Haumann, F. A., and Raphael, M. N.: Atmospheric influences on the anomalous 2016 Antarctic sea ice decay, The Cryosphere, 12, 1103–1119, https://doi.org/10.5194/tc-12-

1103-2018, 2018.

Johnson, S. J., Stockdale, T. N., Ferranti, L., Balmaseda, M. A., Molteni, F., Magnusson, L., Tietsche, S., Decremer, D., Weisheimer, A., Balsamo, G., Keeley, S. P. E., Mogensen, K., Zuo, H., and Monge-Sanz, B. M.: SEAS5: the new ECMWF seasonal forecast system, Geoscientific Model Development, 12, 1087–1117, https://doi.org/10.5194/gmd-12-1087-2019, 2019.

Thépaut, J.-N., Dee, D., Engelen, R., and Pinty, B.: The Copernicus Programme and its Climate Change Service, in: IGARSS 2018 - 2018 IEEE International Geoscience and Remote Sensing Symposium, IGARSS 2018 - 2018 IEEE International Geoscience and Remote Sensing Symposium, 1591–1593, https://doi.org/10.1109/IGARSS.2018.8518067, 2018.

Turner, J., Phillips, T., Marshall, G. J., Hosking, J. S., Pope, J. O., Bracegirdle, T. J., and Deb, P.: Unprecedented springtime retreat of Antarctic sea ice in 2016, Geophysical Research Letters, 44, 6868–6875, https://doi.org/10.1002/2017GL073656, 2017.

Turner, J., Holmes, C., Caton Harrison, T., Phillips, T., Jena, B., Reeves-Francois, T., Fogt, R., Thomas, E. R., and Bajish, C. C.: Record Low Antarctic Sea Ice Cover in February 2022, Geophysical Research Letters, 49, e2022GL098904, https://doi.org/10.1029/2022GL098904, 2022.

Wang, G., Hendon, H. H., Arblaster, J. M., Lim, E.-P., Abhik, S., and van Rensch, P.: Compounding tropical and stratospheric forcing of the record low Antarctic sea-ice in 2016, Nat Commun, 10, 13, https://doi.org/10.1038/s41467-018-07689-7, 2019.

Wang, J., Luo, H., Yang, Q., Liu, J., Yu, L., Shi, Q., and Han, B.: An Unprecedented Record Low Antarctic Sea-ice Extent during Austral Summer 2022, Adv. Atmos. Sci., 39, 1591–1597, https://doi.org/10.1007/s00376-022-2087-1, 2022.

*"sea ice concentration" or area or extent needs mentioning in the title.*

As suggested by the reviewer, we modified the title to "Extended seasonal prediction of Antarctic sea ice concentration using ANTSIC-UNet".

*L 9 the changes to Antarctic Sea Ice a subtle and require more than this introductory sentence – after a period of increasing summer minima there have then been reductions.*

*The Abstract needs more description on why Antarctic sea ice needs predicting. L 15 – 20 can be removed as this is too much detail for an abstract. The final 5 lines are ok as a summary. Some contextualization amongst previous publications is needed for the abstract too.*

Thank you for your comments. We modified the abstract to emphasize the subtle change of Antarctic sea ice and the importance of accuracy prediction.

Antarctic sea ice has experienced rapid change in recent years, with the total sea ice extent abruptly decreasing after a period of gradual increase from the late 1970s until 2014. Accurate long-term predictions of Antarctic sea ice concentration are crucial to support expanding

activities in the Southern Ocean, such as scientific research, tourism and fisheries management. However, dynamic models often face difficulties in accurately predicting Antarctic sea ice due to limited representations of air-ice-sea interactions, especially on seasonal timescales and during the summer months. In response to these challenges, we develop a deep learning model (named ANTSIC-UNet) trained by physically enriched climate variables and evaluate its skill for extended seasonal prediction of Antarctic sea ice concentration (up to 6 months in advance). We compare the predictive skill of ANTSIC-UNet in the Pan- and regional Antarctic with two benchmark models (linear trend and anomaly persistence models). In terms of root-mean-square error (RMSE) for sea ice concentration and integrated ice-edge error (IIEE), ANTSIC-UNet shows much better skills for the extended seasonal prediction, especially for the extreme events in recent years, relative to the two benchmark models. The predictive skill of ANTSIC-UNet is season and region dependent. Sea ice prediction errors increase as lead times increase, with smaller errors observed during autumn and winter, and larger errors in summer. The Pacific and Indian Ocean regions show accurate prediction performance at the sea ice edge during summer. ANTSIC-UNet also shows high predictive skill in capturing the interannual variability of Pan-Antarctic and regional sea ice extent anomalies. We also quantify variable importance through a post-hoc interpretation method. It suggests in addition to sea ice conditions, the ANTSIC-UNet prediction at short lead times shows sensitivity to sea surface temperature, radiative flux, and atmospheric circulation. At longer lead times, zonal wind in the stratosphere appears to be an important influencing factor for the prediction.

*L 25 a first general sentence on the nature of sea ice will help here.*

As suggested by the reviewer, we added the sentence "Sea ice, which formed entirely in the ocean, affects the climate system through modulating the exchange of heat, momentum, moisture and gases between the atmosphere and ocean."

*L 27 this is only true for the summer minimum.*

We modified the sentence "The summer total Antarctic sea ice extent (SIE) has gradually increased until 2014 since the late 1970s and then abruptly decreased."

*L 29, variability in what? I guess extent?*

Yes, we modified the sentence "Antarctic sea ice extent shows large seasonal and interannual variability, and its trend is spatially heterogeneous."

*L 32 everything here after "like" is very vague and needs rewriting.*

Thanks for your comment. We rewrote the sentence "Compared to the Arctic, the prediction of Antarctic sea ice has received much less attention. Also, the demand for subseasonal to extended seasonal Antarctic sea ice predictions has been recognized due to the expanding range of activities in the Southern Ocean (Zampieri et al., 2019; Bushuk et al., 2021; Libera et al., 2022). Accurate sea ice concentration predictions can provide early warnings about sea ice changes and related hazards. This is particularly important for managing the risks of shipping activities in the Southern Ocean. For example, two polar vessels, Akademik Shokalskiy and Xuelong became trapped in rapidly formed sea ice in the Antarctic coastal region (Wang et al., 2014). Commercial fishing and tourism operations mostly use ice-strengthened vessels rather

than icebreakers, which are vulnerable to sea ice hazards. It also supports ecosystem management and informs policy decisions, since the seasonal variations in Antarctic sea ice have a profound influence on marine productivity and fisheries (Libera et al., 2022)."

*L 37 these air-ice-sea interaction processes need further description.*

Thanks for your comment. We added further elaboration of air-ice-sea interaction processes. "Dynamically, sea ice movement and deformation are driven by wind and ocean currents. Thermodynamically, sea ice melting and formation are influenced by convection associated with ocean vertical mixing, heat exchange driven by surface radiation budget and turbulence, and heat advection through horizontal transport of air and water masses"

*L 64 this sentence is difficult to follow. Are linear monthly trends extrapolated to future dates used as a model input?*

Thanks for your comment. The linear monthly trends are extrapolated to future dates and are also used as input to the model. We agree that this sentence was not very clear, therefore we modified this sentence "A linear least-squares trend was fit to observed SIC over the past 30 years at each grid cell for each calendar month and used to predict SIC values for the corresponding month in the following year. These SIC predictions from the linear trend model are also used as the input of ANTSIC-UNet."

*L 66 a description of why reanalysis data is sought is required either here or in an earlier description of the project incentives. What do each data represent and why are they needed for predictions?*

Thanks for your comments. We added the text at the beginning of L66 to clarify the reason for using reanalysis data. "Long-term observations are scarce in the Antarctic, which cannot provide the comprehensive and consistent three-dimensional and time-evolving gridded field of atmosphere and ocean parameters necessary to understand sea ice changes. Reanalysis datasets, which assimilate observations and satellite data, are valuable tools for investigating climate changes in polar regions, offering complete and multivariate descriptions of atmospheric and oceanic conditions."

We added a more detailed description of ERA5 and ORAS5 to explicitly state what each dataset represents and why they are essential for Antarctic sea ice predictions. "ECWMF Reanalysis v5 (ERA5, Hersbach et al., 2020) provides high-resolution and three-dimensional gridded data of comprehensive atmospheric variables from 1940 to the present. ERA5 and its predecessor ERA-Interim are widely regarded as the best-performing reanalysis datasets in polar regions, with particularly reliable analyses over the Southern Ocean compared with surface and upper-level observations (Bracegirdle & Marshall, 2012; Bromwich et al., 2011). Ocean Reanalysis System 5 (ORAS5, Zuo et al., 2019) is a global eddy-permitting ocean and sea-ice ensemble reanalysis which provides historical ocean and sea-ice conditions from 1979 to the present, which adopts the same sea surface temperature as ERA5 taken from observations. Sea ice changes are strongly influenced by the atmosphere above and the ocean below through dynamic and thermodynamic processes. Therefore, the relevant atmospheric variables selected from ERA5 and oceanic variables obtained from ORAS5 are also used as inputs by ANTSIC-UNet to investigate the key factors contributing to sea ice predictions in the complex interaction between sea ice, ocean and atmosphere."

*L 76 Is the input data volume held static throughout all development? The data lag is often an option that requires testing and investigation.*

Yes, the input data volume was static throughout all development. The variable importance analysis helped identify the most effective combination of relevant variables at different time lags to enhance prediction accuracy. In future research, we plan to investigate the impact of the time length of individual climate variables by retraining the deep learning model. This will allow us to assess how such changes in data lags affect the model's predictive performance, though it will require significant computational resources.

*L 72 why is v10hPa not included also?*

Thank you for your query regarding the exclusion of v10hPa. Other studies have already clarified that the changes in stratospheric zonal circulation predominantly affect the circumpolar westerly winds in the troposphere through downward propagation, which in turn affects the sea ice distribution and variability (Wang et al., 2019; Cordero et al., 2023). Therefore, we only include 10-hPa zonal wind.

Reference:

Cordero, R. R., Feron, S., Damiani, A., Llanillo, P. J., Carrasco, J., Khan, A. L., Bintanja, R., Ouyang, Z., and Casassa, G.: Signature of the stratosphere–troposphere coupling on recent record-breaking Antarctic sea-ice anomalies, The Cryosphere, 17, 4995–5006, https://doi.org/10.5194/tc-17-4995-2023, 2023.

Wang, G., Hendon, H. H., Arblaster, J. M., Lim, E.-P., Abhik, S., and van Rensch, P.: Compounding tropical and stratospheric forcing of the record low Antarctic sea-ice in 2016, Nat Commun, 10, 13, https://doi.org/10.1038/s41467-018-07689-7, 2019.

*L 108 The linear trend prediction is not described well in section 2.1*

Thanks for your comment. We agree that this statement was not very clear. We rewrote the description of the linear trend prediction as "A linear least-squares trend was fit to observed SIC over the past 30 years at each grid cell for each calendar month and used to predict SIC values for the corresponding month in the following year. These SIC predictions from the linear trend model are also used as the input of ANTSIC-UNet."

*L 112 This implies that the RHS of equation 1 is just the observed ice concentration field. What benefit is this? Further description of how anomaly persistence works as a prediction is needed here.*

The benefit of the anomaly persistence model lies in its straightforward application to give a continuous prediction of the variable by carrying forward the initial state of anomalies. This statistical method has been widely used as a benchmark for predicting sea ice concentration on seasonal timescales since sea ice conditions often change gradually rather than abruptly (Wayand et al., 2019; Bushuk et al., 2021; Niraula and Goessling, 2021). The effectiveness of the anomaly persistence decreases with increasing lead time as the influence of initial anomalies diminishes.

We updated the explanation in the text "The anomaly persistence works by preserving the deviations from the climatological anomalies and assuming these anomalies will persist into the future. For example, if a particular region currently has more sea ice than average, this positive anomaly will continue as time increases. This statistical method has been widely used as a benchmark for predicting sea ice concentration on seasonal timescales since sea ice conditions often change gradually rather than abruptly (Wayand et al., 2019; Bushuk et al., 2021; Niraula and Goessling, 2021). While this method is effective for short-term forecasts, its accuracy declines over longer lead times as the influence of initial anomalies weakens."

Reference:

Bushuk, M., Winton, M., Haumann, F. A., Delworth, T., Lu, F., Zhang, Y., Jia, L., Zhang, L., Cooke, W., Harrison, M., Hurlin, B., Johnson, N. C., Kapnick, S. B., McHugh, C., Murakami, H., Rosati, A., Tseng, K.-C., Wittenberg, A. T., Yang, X., and Zeng, F.: Seasonal Prediction and Predictability of Regional Antarctic Sea Ice, Journal of Climate, 34, 6207–6233, https://doi.org/10.1175/JCLI-D-20-0965.1, 2021.

Niraula, B. and Goessling, H. F.: Spatial Damped Anomaly Persistence of the Sea Ice Edge as a Benchmark for Dynamical Forecast Systems, Journal of Geophysical Research: Oceans, 126, e2021JC017784, https://doi.org/10.1029/2021JC017784, 2021.

Wayand, N. E., Bitz, C. M., and Blanchard-Wrigglesworth, E.: A Year-Round Subseasonal-to-Seasonal Sea Ice Prediction Portal, Geophysical Research Letters, 46, 3298–3307, https://doi.org/10.1029/2018GL081565, 2019.

*L 165 key acronyms need defining in each figure caption. (and all others too)*

Thank you for your comment. We updated all relevant figure and table captions to include definitions for acronyms such as RMSE (root-mean-square error) and IIEE (integrated ice-edge error), ensuring that readers can easily understand the terms used without needing to look back to the main text. For example, we modified the caption for Table. 2 as follows:

"Table 2. The averaged predictive skill of Antarctic sea ice for ANTSIC-UNet, linear trend and anomaly persistence models for all testing years (RMSE: root-mean-square error; IIEE: integrated ice-edge error)"

*L 253 "extremely low" rephrase with better accuracy.*

*L 253 this table needs extra columns to show what was extreme about these years – SIE/SIC anomalies perhaps.*

Thank you for your comments. We modified the title of the table and added the extra columns.

Table 3. The averaged predictive skill of ANTSIC-UNet, linear trend and anomaly persistence models for the extreme summer years of Antarctic sea ice extent. Here, Observed SIEA represents February monthly anomalies of sea ice extent from NSIDC observations for these extreme years, calculated by subtracting the February average sea ice extent for the period 1981-2011 (units: million square kilometers). RMSE: root-mean-square error; IIEE: integrated ice-edge error.

|  | Observed SIEA | Metrics | ANTSIC-UNet | Linear trend | Anomaly persistence |
|---|---|---|---|---|---|
| 2017 | -0.76 | RMSE | 0.21 | 0.25 | 0.24 |
|  |  | IIEE | 1.80 | 2.56 | 2.52 |
| 2022 | -0.84 | RMSE | 0.21 | 0.22 | 0.23 |
|  |  | IIEE | 1.68 | 2.24 | 2.45 |
| 2023 | -1.14 | RMSE | 0.24 | 0.27 | 0.31 |
|  |  | IIEE | 2.00 | 3.05 | 3.11 |

---

## Referee Report (RR1)

**Review: Extended seasonal prediction of Antarctic sea ice concentration using ANTSIC-UNet**

This is a well written paper exploring the development and application of a convolutional neural network (CNN), ANTSIC-UNet, for seasonal predictions of Antarctic sea ice concentration (SIC). The paper demonstrates how ANTSIC-UNet outperforms two benchmark models, as well as the SEAS5 numerical sea ice forecasting model. This paper also explores variable importance via the use of the explainable AI tool, permute and predict. I recommend this paper for publication subject to the major revisions outlined below.

Whilst the research presented here is of high quality, the abstract, introduction and discussion need to emphasise the novelty brought by this paper. This is currently not clear to the reader. The introduction highlights that fewer studies have predicted SIC in the Antarctic compared with the Arctic. Although the application of sea ice forecasts to the Antarctic provides some novelty, greater clarification of the methodological novelty provided by this study is also required. For example, previous studies have already applied CNNs for sea ice forecasting, undertaken analysis of feature importance and compared ML model outputs to SEAS5. This clarification of methodological novelty will make it easier for the reader to follow the paper.

Linked to this point, whilst it is true that far fewer papers have forecast Antarctic SIC, some key publications are missing. Please cite these and contextualise the findings of this paper to these manuscripts:

Dong, X., Yang, Q., Nie, Y., Zampieri, L., Wang, J., Liu, J. and Chen, D., 2024. Antarctic Sea Ice Prediction with A Convolutional Long Short-Term Memory Network. *Ocean Modelling*, p.102386.

Lin, Y., Yang, Q., Li, X., Dong, X., Luo, H., Nie, Y., Wang, J., Wang, Y. and Min, C., 2025. Ice-kNN-South: A lightweight machine learning model for Antarctic sea ice prediction. *Journal of Geophysical Research: Machine Learning and Computation*, *2*(1), p.e2024JH000433.

Wang, Y., Yuan, X., Ren, Y., Bushuk, M., Shu, Q., Li, C. and Li, X., 2023. Subseasonal prediction of regional Antarctic sea ice by a deep learning model. *Geophysical Research Letters*, *50*(17), p.e2023GL104347.

As a general point, I also believe this paper requires some restructuring. The comparison of ANTSIC-UNet to SEAS5 is not mentioned until the discussion section on line 395. The use of SEAS5 requires mentioning in the introduction, methods, and results. Further, it is not common for the discussion section to provide new results and figures. I suggest Figures 10, 11 and 12, alongside the supporting text and equations describing these results, are moved to the results section. This will allow the discussion section to focus on the relevance and contextualisation of the results, making the paper easier to follow for the reader.

Overall, the paper reads very well with very few typographical errors, I suggest these further minor corrections:

Line 77 - 79: please provide detail on the algorithm used to convert from passive microwave brightness temperatures to sea ice concentration values.

Section 2.2. Please justify the use of a CNN with UNet architecture. Some recent papers have shown generative models or other AI approaches to outperform UNets. Were other ML algorithms and architectures considered?

Line 81 – 84: "A linear least-squares trend was fit...." This information does not fit under the subsection 2.1, as these lines are describing a method applied to the passive microwave data, rather than the data itself. Please create a new subsection in the methods section on the benchmark models.

Line 95 - 105: Please refer to Table 1 when listing all the variables.

Line 110: Please make clear here or somewhere else the temporal resolution of the forecasts. Is it monthly, daily, seasonal or some other resolution?

Line 135: Please describe the hyperparameter selection and tuning process you employed.

Figure 2: Please provide some background in the introduction on how and why the Southern Ocean is split into these five regions.

Figure 2 caption: typo: "based on the same calendat month".

Figure 3 a) and e), please flip the colour ramp so white is ice and water is blue, or use a separate colour ramp altogether. Same for Figure 7.

Figure 3: Please make clear whether these are February and September means for a particular year or the whole date range.

Figure 4 f1 and f2: please make clear that A and B stand for Amundsen and Bellingshausen.

Discussion section: Due to the large similarities between this paper and the IceNet model published in Andersson et al. (2021), please provide detailed discussion on the relative performance of ANTSIC-UNet and IceNet.

Figure 6 line 262-263: "ANTSIC-UNet (anomaly persistent model) at different lead times up to 6 months...." How are different lead times represented in this figure?

Section 3.3: As a general discussion point, what is the suitability of using the anomaly persistence model as a benchmark model for forecasting extreme events? Isn't it always destined to underrepresent these extremely anomalous cases? Are there more appropriate benchmark models that could be used for these circumstances?

Table 3 line 298: Typo "Here, **O**bserved.." change to lower case.

Figure 8: Help the reader- in which month when did the extreme event(s) occur.

Section 3.4. There are some points that are more appropriate for the discussion, particularly where references are made to other papers. For example from line 320 "previous studies....."

Line 334: Typo: Circulation. (Raphael...) – remove full stop.

Discussion: Please contextualise your findings on feature importance for sea ice forecasting with the following paper that also carried out a similar study:

Uebbing, L., Joakimsen, H.L., Luppino, L.T., Martinsen, I., McDonald, A., Wickstrøm, K.K., Lefèvre, S., Salberg, A.B., Hosking, S. and Jenssen, R., 2025, January. Investigating the Impact of Feature Reduction for Deep Learning-based Seasonal Sea Ice Forecasting. In *Northern Lights Deep Learning Conference 2025*.

Discussion: Why does the performance differ between the different regions of the Southern Ocean? For example, the disparities between  $4 \, b1 - f1$ . There is mention of this on lines 367 - 369, but please expand further. Also, please comment on the better predictive performance of the tool in the Austral summer.

There is no conclusion section. Please check if this is required.

---

## Author Response (AR2)

**Response to comments by Reviewer #1**

We would like to thank the reviewer for the helpful comments on the manuscript. Please find below our responses to the comments.

The authors have addressed my previous comments. The ms is clearly improved. I still have a few editorial comments.

1. In the comparison of predictive skill between ANTSIC-UNet and HIS-V, how does ANTSIC-UNet perform in terms of ACC?

Thanks for this question and concern about the ACC of DL models. Figure R1 shows the ACC of HIS-V trained by historical data without incorporating the future 6 months linear trend predictions of SIC, and the difference in ACC between HIS-V and ANTSIC-UNet. For the Pan-Antarctic, ANTSIC-UNet shows higher ACC from February to July and October to December at short lead times, and lower ACC as lead time increases, with contributions from all five sectors. Specifically, lower ACC is found in the Weddell Sea, Indian Ocean, and Pacific Ocean from December to April as the lead time exceeds 3 months. Higher ACC is observed in the Ross Sea from January to March, and the Amundsen and Bellingshausen Seas show a broad coverage of relatively high ACC. Additionally, the Pacific Ocean consistently exhibits higher ACC from July to September across all lead times. These differences in the interannual variability of SIE anomalies may be linked to the different inherent sea ice trends in these regions. For instance, the Indian Ocean experiences significant interannual fluctuations, with total sea ice area reaching its maximum in October 2010, followed by a decline to a record low in 2016, and subsequent recovery. Therefore, incorporating the linear trend prediction of SIC may reduce the predictive performance of the deep learning model in most seasons of the Indian Ocean. Furthermore, when incorporating the linear trend predictions of SIC and considering the interactions between sea ice and other climate variables, the ANTSIC-UNet shows improved skill in capturing the interannual variability of SIE anomalies throughout the year in the Ross Sea, Amundsen and Bellingshausen Seas, and during summer in the Pacific Ocean.

Figure R1. The ACC (a1-f1) between the observed and HIS-V (DL model trained by historical data, without incorporating the future 6 months linear trend predictions of sea ice concentration) predicted regional SIE anomalies for different target months and forecast lead times during 1981-2023. (a2-f2) as (a1-f1) but for the ACC difference between HIS-V and ANTSIC-UNet.

2. The authors use the permutation feature-importance method to explain model variance, which is primarily based on the distance between predictions and observations. Given this, are the variable importances consistent with RMSE in terms of the prediction of the SIC variability?

Yes, the permutation feature importance is consistent with the RMSE of SIC spatial variability. We quantify the importance of each variable by calculating the change in the model evaluation metric (RMSE between the predicted SIC by the trained model and observed SIC) before and after permuting the particular variable.

**Response to comments by Reviewer #2**

We would like to thank the reviewer for the helpful comments on the paper. Please find below our responses to the comments.

This is a well written paper exploring the development and application of a convolutional neural network (CNN), ANTSIC-UNet, for seasonal predictions of Antarctic sea ice concentration (SIC). The paper demonstrates how ANTSIC-UNet outperforms two benchmark models, as well as the SEAS5 numerical sea ice forecasting model. This paper also explores variable importance via the use of the explainable AI tool, permute and predict. I recommend this paper for publication subject to the major revisions outlined below.

Whilst the research presented here is of high quality, the abstract, introduction and discussion need to emphasise the novelty brought by this paper. This is currently not clear to the reader. The introduction highlights that fewer studies have predicted SIC in the Antarctic compared with the Arctic. Although the application of sea ice forecasts to the Antarctic provides some novelty, greater clarification of the methodological novelty provided by this study is also required. For example, previous studies have already applied CNNs for sea ice forecasting, undertaken analysis of feature importance and compared ML model outputs to SEAS5. This clarification of methodological novelty will make it easier for the reader to follow the paper. Linked to this point, whilst it is true that far fewer papers have forecast Antarctic SIC, some key publications are missing. Please cite these and contextualise the findings of this paper to these manuscripts:

Dong, X., Yang, Q., Nie, Y., Zampieri, L., Wang, J., Liu, J. and Chen, D., 2024. Antarctic Sea Ice Prediction with A Convolutional Long Short-Term Memory Network. Ocean Modelling, p.102386.

Lin, Y., Yang, Q., Li, X., Dong, X., Luo, H., Nie, Y., Wang, J., Wang, Y. and Min, C., 2025. Ice-kNN-South: A lightweight machine learning model for Antarctic sea ice prediction. Journal of Geophysical Research: Machine Learning and Computation, 2(1), p.e2024JH000433.

Wang, Y., Yuan, X., Ren, Y., Bushuk, M., Shu, Q., Li, C. and Li, X., 2023. Subseasonal prediction of regional Antarctic sea ice by a deep learning model. Geophysical Research Letters, 50(17), p.e2023GL104347.

Thank you for your comment. We have revised the abstract, introduction and discussion to more clearly emphasize the novelty and contributions of our study. In the abstract and discussion, we have clarified the distinctions and advantages of our deep learning model compared to previous studies on Antarctic sea ice prediction. Specifically, we address key challenges in both deep learning models and dynamical models, particularly their limited representation of air-ice-sea interactions and lack of interpretability, by training our deep learning model (ANTSIC-UNet) using multiple climate variables. In addition, we explore the relative importance of these variables using the permutation feature importance approach to enhance the interpretability of our model. Moreover, we place significant emphasis on the model's performance for extended seasonal predictions (i.e., longer lead times) and conduct a systematic evaluation during extreme sea-ice years, which have both received little attention in previous studies.

In the introduction, we have added the recent key publications on Antarctic SIC prediction, as highlighted by the reviewer, to provide a comprehensive context for our work. For example, we revised the introduction as follows:

Original: "Recently, Wang et al. (2023) developed a SIPNet model with encoder-decoder structure for subseasonal Antarctic sea ice concentration prediction, which outperforms some dynamical models and advanced linear statistical models. Nevertheless, these DL methods were trained by pure historical sea ice concentration data without considering underlying physical processes governing the variation of Antarctic sea ice."

Revised: "Recently, Wang et al. (2023) developed a SIPNet model with encoder-decoder structure for subseasonal Antarctic sea ice concentration prediction, which outperforms some dynamical models and advanced linear statistical models at lead times of 1-8 weeks. Dong et al. (2024) employed a convolutional long short-term memory (ConvLSTM) network to predict Antarctic SIC up to 60 days ahead, which shows skillful predictions within 30 days and accurately forecasts annual maximum and minimum sea ice extents from 2017 to 2022. However, ConvLSTM demands significant computational resources during training, and relies on iterative forecasting which leads to error accumulation over time and requires a trade-off between accuracy and prediction length. Lin et al. (2025) proposed Ice-KNN-South, a lightweight machine learning model for predicting daily Antarctic SIC at lead times of 1-90 days. While these studies have made significant contributions, they primarily rely on historical

SIC data without considering underlying physical processes governing the variation of Antarctic sea ice. Furthermore, they focus on shorter prediction horizons, and their skillfulness in extended seasonal forecasting remains unknown."

As a general point, I also believe this paper requires some restructuring. The comparison of ANTSIC-UNet to SEAS5 is not mentioned until the discussion section on line 395. The use of SEAS5 requires mentioning in the introduction, methods, and results. Further, it is not common for the discussion section to provide new results and figures. I suggest Figures 10, 11 and 12, alongside the supporting text and equations describing these results, are moved to the results section. This will allow the discussion section to focus on the relevance and contextualisation of the results, making the paper easier to follow for the reader.

Thank you for your comment. We agree that reorganizing the paper would improve its clarity and flow. In response to the comment, we have moved the introduction of SEAS5 to the methods section. We now include the comparison of ANTSIC-UNet with the statistical models (a linear trend model and an anomaly persistence model) and a dynamical model (SEAS5) in the results section. Additionally, we have added a new Section 3.5: "Physical constraints" to describe the results with the incorporation of physical constraints into ANTSIC-UNet. Finally, we have revised the abstract and discussion sections to align with these changes. We hope this restructuring addresses the reviewer's concern.

Overall, the paper reads very well with very few typographical errors, I suggest these further minor corrections:

Line 77 - 79: please provide detail on the algorithm used to convert from passive microwave brightness temperatures to sea ice concentration values.

Thank you for your comment. The monthly SIC data are derived using the Bootstrap algorithm, which utilizes brightness temperature observations from the 37H, 37V, and 19V channels to estimate sea ice concentration (Comiso et al., 1997; Comiso and Nishio, 2008). We modified the sentence as follows:

Original: "In this study, monthly Antarctic sea ice concentration (SIC) data obtained from the National Snow and Ice Data Center (NSIDC) (https://nsidc.org/data/nsidc-0079/versions/3) are used as the input of ANTSIC-UNet, and are derived from brightness temperature of the

Scanning Multichannel Microwave Radiometer (SMMR), the Special Sensor Microwave/Imager (SSM/I) sensors, and the Special Sensor Microwave Imager/Sounder (SSMIS)."

Revised: "In this study, monthly Antarctic sea ice concentration (SIC) data obtained from the National Snow and Ice Data Center (NSIDC) (https://nsidc.org/data/nsidc-0079/versions/3) are used as the input of ANTSIC-UNet, and are derived from brightness temperature of the Scanning Multichannel Microwave Radiometer (SMMR), the Special Sensor Microwave/Imager (SSM/I) sensors, and the Special Sensor Microwave Imager/Sounder (SSMIS). SIC is retrieved using the Bootstrap algorithm, which utilizes brightness temperature observations from the 37H, 37V, and 19V channels to estimate sea ice concentration (Comiso et al., 1997; Comiso and Nishio, 2008)."

**Reference:**

Comiso, J. C., Cavalieri, D. J., Parkinson, C. L., and Gloersen, P.: Passive microwave algorithms for sea ice concentration: A comparison of two techniques, Remote Sensing of Environment, 60, 357–384, https://doi.org/10.1016/S0034-4257(96)00220-9, 1997.

Comiso, J. C. and Nishio, F.: Trends in the sea ice cover using enhanced and compatible AMSR-E, SSM/I, and SMMR data, Journal of Geophysical Research: Oceans, 113, https://doi.org/10.1029/2007JC004257, 2008.

Section 2.2. Please justify the use of a CNN with UNet architecture. Some recent papers have shown generative models or other AI approaches to outperform UNets. Were other ML algorithms and architectures considered?

Thank you for your comment. The primary goal of this study was to explore the feasibility of using complex climate variables to predict Antarctic sea ice concentration (SIC) and to investigate the interpretability of deep learning models. The fully convolutional neural network (FCN) based on the U-Net architecture, known for its simplicity and effectiveness in handling spatial data, was chosen as a useful tool to achieve this objective. Our results show that ANTSIC-UNet based on a relatively simple U-Net architecture, can effectively capture the complex relationships between climate variables and sea ice dynamics, and outperform benchmark models and state-of-the-art dynamical models (e.g., SEAS5) in predicting Antarctic

sea ice.

While generative models, such as Generative Adversarial Networks (GANs), have shown promise in certain applications, they often require significantly more computational resources and training time. Given the exploratory nature of this study and the need for efficient experimentation, we opted for the U-Net architecture, which strikes a balance between performance and computational efficiency. In future work, we plan to explore the use of these AI-based models to assess whether they can provide additional predictive improvements. We will also continue to investigate the interpretability of these models and their ability to incorporate physical constraints to advance our understanding of Antarctic sea ice change.

Line 81 - 84: "A linear least-squares trend was fit...." This information does not fit under the subsection 2.1, as these lines are describing a method applied to the passive microwave data, rather than the data itself. Please create a new subsection in the methods section on the benchmark models.

Thank you for your comment. We agree that the information in lines 81–84 should be moved to a more appropriate section. As suggested, we have removed these sentences and created a new subsection (Section 2.3) to describe the benchmark models, and subsequent section numbers have been updated accordingly to maintain the proper structure.

**"2.3 Benchmark models**

In this study, the linear trend and anomaly persistence predictions are used as benchmarks to assess the predictive skill of ANTSIC-UNet. The linear trend model involves fitting a linear least-squares trend to observed SIC over the past 30 years at each grid cell for each calendar month. This trend is then used to predict SIC values for the corresponding calendar month in the following year. Additionally, these SIC predictions from this linear trend model are also used as the input to ANTSIC-UNet.

The anomaly persistence prediction is calculated as follows:

$$SIC_{pred}(t+\tau) = SIC_{clim}(t+\tau) + SIC_{anom}(t)$$
 (1)

where  $SIC_{pred}$  is the target month predicted ice concentration at the lead time  $\tau$ ,  $SIC_{clim}$  is the climatogy ice concentration at the target month, and  $SIC_{anom}$  is the observed ice concentration anomaly relative to the climatology at the initial time. The climatology for each

month is computed for the period of the training data (1979-2011). The anomaly persistence works by preserving the deviations from the climatological anomalies and assuming these anomalies will persist into the future. For example, if a particular region currently has more sea ice than average, this positive anomaly will continue as time progresses. This statistical method has been widely used as a benchmark for predicting sea ice concentration on seasonal timescales, since sea ice conditions often change gradually rather than abruptly (Wayand et al., 2019; Bushuk et al., 2021; Niraula and Goessling, 2021). While this method is effective for short-term forecasts, its accuracy declines over longer lead times as the influence of initial anomalies weakens."

Line 95 - 105: Please refer to Table 1 when listing all the variables.

Thank you for your comment. We revised the MS to refer to Table 1 when listing all the variables.

Original: "These variables include 2m air temperature (T2), 500-hPa air temperature (T500), sea surface temperature (SST), ocean temperature (PT), ocean heat content for the upper 300m (OHC300), downwelling solar radiation (DSR), upwelling solar radiation (USR), sea level pressure (SLP), 500-hPa geopotential height (H500), 250-hPa geopotential height (H250), 10m u-component of wind (U10), 10m v-component of wind (V10), and 10-hPa zonal wind (U10hPa)."

Revised: "These variables are listed in Table 1 and include 2m air temperature (T2), 500-hPa air temperature (T500), sea surface temperature (SST), ocean temperature (PT), ocean heat content for the upper 300m (OHC300), downwelling solar radiation (DSR), upwelling solar radiation (USR), sea level pressure (SLP), 500-hPa geopotential height (H500), 250-hPa geopotential height (H250), 10m u-component of wind (U10), 10m v-component of wind (V10), and 10-hPa zonal wind (U10hPa)."

Line 110: Please make clear here or somewhere else the temporal resolution of the forecasts. Is it monthly, daily, seasonal or some other resolution?

Thank you for your comment. To clarify the temporal resolution of the forecasts, we revised the sentence as follows:

Original: "The final output provides the 6-month forecast of Antarctic sea ice concentration."

Revised: "The final output provides the 6-month forecast of monthly Antarctic sea ice concentration."

Line 135: Please describe the hyperparameter selection and tuning process you employed.

Thank you for your comment. We added the description of the hyperparameter selection and tuning process as follows:

"Here, we use typical hyperparameters for the deep learning model. The kernel size for the convolutional layers is set to (3,3). Due to memory constraints, we set the batch size to 2. The loss function applied is the mean squared error (MSE), with a learning rate of 0.0001 and a weight decay of 0. The Adam optimizer is used for training."

Figure 2: Please provide some background in the introduction on how and why the Southern Ocean is split into these five regions.

Thank you for your comment. We added more details in the introduction regarding the division of the Southern Ocean into five sectors as follows:

"Sea ice in different regions exhibits complex spatial patterns of change in growth, retreat, and duration (Liang et al., 2023). The Southern Ocean sea ice region is divided into five sectors: the Weddell Sea, Indian Ocean, Pacific Ocean, Amundsen and Bellingshausen Seas, and Ross Sea. These regions are characterised by their unique climatic, oceanographic, and geographical characteristics (Zwally et al., 2002; Grieger et al., 2018; Josey et al., 2024). This division has been widely used in studying the regional dynamics and prediction of Antarctic sea ice (e.g., Eayrs et al., 2019; Bushuk et al., 2021; Liang et al., 2023)."

**Reference:**

Bushuk, M., Winton, M., Haumann, F. A., Delworth, T., Lu, F., Zhang, Y., Jia, L., Zhang, L., Cooke, W., Harrison, M., Hurlin, B., Johnson, N. C., Kapnick, S. B., McHugh, C., Murakami, H., Rosati, A., Tseng, K.-C., Wittenberg, A. T., Yang, X., and Zeng, F.: Seasonal Prediction and

Predictability of Regional Antarctic Sea Ice, Journal of Climate, 34, 6207–6233, https://doi.org/10.1175/JCLI-D-20-0965.1, 2021.

Eayrs, C., Holland, D., Francis, D., Wagner, T., Kumar, R., and Li, X.: Understanding the Seasonal Cycle of Antarctic Sea Ice Extent in the Context of Longer-Term Variability, Reviews of Geophysics, 57, 1037–1064, https://doi.org/10.1029/2018RG000631, 2019.

Grieger, J., Leckebusch, G. C., Raible, C. C., Rudeva, I., and Simmonds, I.: Subantarctic cyclones identified by 14 tracking methods, and their role for moisture transports into the continent, Tellus A: Dynamic Meteorology and Oceanography, 70, 2018.

Liang, K., Wang, J., Luo, H., and Yang, Q.: The Role of Atmospheric Rivers in Antarctic Sea Ice Variations, Geophysical Research Letters, 50, e2022GL102588, https://doi.org/10.1029/2022GL102588, 2023.

Zwally, H. J., Comiso, J. C., Parkinson, C. L., Cavalieri, D. J., and Gloersen, P.: Variability of Antarctic sea ice 1979–1998, Journal of Geophysical Research: Oceans, 107, 9-1-9–19, https://doi.org/10.1029/2000JC000733, 2002.

Figure 2 caption: typo: "based on the same calendat month".

The typo has been corrected.

Figure 3 a) and e), please flip the colour ramp so white is ice and water is blue, or use a separate colour ramp altogether. Same for Figure 7.

Thank you for your comment. We modified the figures and used a separate colour ramp. Additionally, we included the SEAS5 predictions for comparison.

Figure R2. The monthly mean sea ice concentration of the NSIDC observations for (a) February and (f) September, and the errors in predicting by ANTSIC-UNet (b1-b3, g1-g3), the linear trend model (c and h), anomaly persistence model (d1-d3, i1-i3) and SEAS5 (e1-e3, j1-j3) at lead time of 1, 3, and 5 months for February (upper panel) and September (lower panel) during the testing years.

Figure R3. February and September 2023 SIC of NSIDC observations (a, e) and errors predicted by ANTSIC-UNet (b1-b3, g1-g3), the linear trend model (c and h), anomaly persistence model (d1-d3, i1-i3) and SEAS5 (e1-e3, j1-j3) at lead time of 1, 3 and 5 months (lowest sea ice extent on record).

Figure 3: Please make clear whether these are February and September means for a particular year or the whole date range.

Thank you for your comment. We indicated in the figure caption that the data represent the mean for February and September for the testing years.

Figure 4 fl and f2: please make clear that A and B stand for Amundsen and Bellingshausen.

Thank you for your comment. We modified the caption of Figure 4 as follows:

Original: "Figure 4. The predictive skill of sea ice concentration (spatially and temporally averaged during the testing years) in terms of RMSE and IIEE (units: million square kilometers) between the ANTSIC-UNet predictions and NSIDC observations for different target months and forecast lead times."

Revised: "Figure 4. The predictive skill of sea ice concentration (spatially and temporally averaged during the testing years) in terms of RMSE and IIEE (units: million square kilometers) between the ANTSIC-UNet predictions and NSIDC observations for different target months and forecast lead times. 'A and B' in (f1) and (f2) refer to the Amundsen Sea and Bellingshausen Seas, respectively."

Discussion section: Due to the large similarities between this paper and the IceNet model published in Andersson et al. (2021), please provide detailed discussion on the relative performance of ANTSIC-UNet and IceNet.

Thank you for your comment. Although ANTSIC-UNet and the IceNet model proposed by Andersson et al. (2021) have similarities in their underlying U-Net architecture, the two models differ in design, objectives, and application domains, making direct comparisons difficult.

IceNet was designed for Arctic sea ice classification, aiming to predict three discrete SIC categories: open water (SIC<=15%), marginal ice (15%<SIC<80%), and full ice (SIC>=80%). In contrast, ANTSIC-UNet is developed for Antarctic sea ice concentration (SIC) regression prediction. This difference in task leads to different loss functions being used during training: classification models like IceNet use categorical loss functions (e.g., cross-entropy), while

ANTSIC-UNet employs regression-based loss functions (e.g., mean squared error) to predict continuous SIC values.

Moreover, the Arctic and Antarctic have different geographical features, which lead to major differences in oceanic and atmospheric circulation patterns (Maksym, 2019). As a result, the sea ice in the Antarctic and Arctic shows completely different trends and behaviors. The Antarctic and Arctic also experience extreme sea ice events in different ways, which are driven by different atmospheric and oceanic factors.

Deep learning models such as IceNet and ANTSIC-UNet exhibit strong nonlinear learning capabilities, which are particularly valuable in predicting extreme events that deviate from climatological norms. Figure R4 in Andersson et al. (2021) shows IceNet's predictive skill for seasonal September forecasts in the Arctic. The 2012–2020 period contains three anomalous September Arctic SIEs: 2012 (lowest extent on record), 2013 (anomalously high extent), and 2020 (second lowest extent on record). IceNet shows skillful predictions for these extreme events, outperforming the linear trend predictions and SEAS5, except in September 2013, when its error slightly exceeded that of SEAS5 at 2-3 months lead. In our study, we placed particular emphasis on evaluating model performance during three extreme summer sea ice events in the Antarctic (2017, 2022 and 2023). As shown in Figure R5, ANTSIC-UNet outperforms SEAS5 and linear trend predictions for sea ice edge error in all extreme summer years. Therefore, despite bring designed for different hemispheres and sea ice prediction tasks, both IceNet and ANTSIC-UNet highlight the strength of deep learning models in capturing nonlinear changes, particularly in extreme sea ice years.

Figure R4. Comparing IceNet with SEAS5 and the linear trend for seasonal September forecasts. a–i IceNet's improvement in binary accuracy relative to SEAS5 and the linear trend models for September forecasts at 4- to 2-month lead times for the validation and test years (2012–2020) (From Andersson et al., 2020; Figure 4).

Figure R5. Integrated ice-edge error (IIEE) of ANTSIC-UNet, the linear trend forecast and SEAS5 for February forecasts at lead time of 1, 3, and 5 months for the extreme summer years. (a) 2017, (b) 2022 and (c) 2023.

Reference:

Maksym, T.: Arctic and Antarctic Sea Ice Change: Contrasts, Commonalities, and Causes, Annual Review of Marine Science, 11, 187–213, https://doi.org/10.1146/annurev-marine-

010816-060610, 2019.

Figure 6 line 262-263: "ANTSIC-UNet (anomaly persistent model) at different lead times up

to 6 months...." How are different lead times represented in this figure?

Thank you for your comment. In Figure 6, the different lead times are represented on the xaxis, which ranges from 1 to 6 months, corresponding to the lead times up to 6 months as

mentioned in the caption of Figure 6.

Section 3.3: As a general discussion point, what is the suitability of using the anomaly

persistence model as a benchmark model for forecasting extreme events? Isn't it always

destined to underrepresent these extremely anomalous cases? Are there more appropriate

benchmark models that could be used for these circumstances?

Thank you for your comment. For shorter lead times, the anomaly persistence model does not

always underestimate such cases due to the inherent characteristics of sea ice. The persistence

of sea ice anomalies can often continue in the short term, errors tend to become more significant

because the anomaly persistence model fails to capture the longer-term variability and more

complex interactions of extreme events as lead time increases. Using more complex benchmark

models, such as multiple linear regression and random forest, which can incorporate additional

predictors (e.g., atmospheric and oceanic variables) and partially capture the nonlinear

relationship, may provide a more appropriate reference for evaluating the predictive ability of

deep learning models.

Table 3 line 298: Typo "Here, Observed.." change to lower case.

The typo has been corrected.

Figure 8: Help the reader- in which month when did the extreme event(s) occur.

Thank you for your comment. We modified the caption for Figure 8 as follows:

17

Original: "Figure 8. Seasonality errors of the Pan- and regional Antarctic monthly mean SIE

(SIC > 15%) between NSIDC observations and ANTSIC-UNet (a-f) and anomaly persistence

model (g-l) predictions at different lead times for 2023 (lowest sea ice extent on record). The

black lines show the seasonality SIE errors between observations and linear trend model. (units:

million square kilometers)"

Revised: "Figure 8. Seasonality errors of the Pan- and regional Antarctic monthly mean SIE

(SIC > 15%) between NSIDC observations and ANTSIC-UNet (a-f) and anomaly persistence

model (g-l) predictions at different lead times for 2023 (lowest sea ice extent on record in

February). The black lines show the seasonality SIE errors between observations and linear

trend model. (units: million square kilometers)"

Section 3.4. There are some points that are more appropriate for the discussion, particularly

where references are made to other papers. For example from line 320 "previous studies....."

Thank you for your comment. Regarding the sentence, "Previous studies suggested that the

evaluation metrics of model's predictive skill, particularly for models with strong

generalization ability, correlate closely with feature importance (FI) (Andersson et al., 2021;

Molnar, 2019)." we would like to clarify that the principle behind the permutation feature

importance method we use is consistent with the method described in the referenced studies.

We intend to retain this content as it provides the necessary background and theoretical support,

helping readers understand how this method has been applied in previous research and its

relevance to our study.

However, to better contextualize our findings in relation to other studies, we expanded on this

in the discussion section. For example, as the reviewer suggested, we associated our feature

importance results with those of Uebbing et al. (2025).

Line 334: Typo: Circulation. (Raphael...) – remove full stop.

The typo has been corrected.

Discussion: Please contextualise your findings on feature importance for sea ice forecasting

with the following paper that also carried out a similar study:

Uebbing, L., Joakimsen, H.L., Luppino, L.T., Martinsen, I., McDonald, A., Wickstrøm, K.K., Lefèvre, S., Salberg, A.B., Hosking, S. and Jenssen, R., 2025, January. Investigating the Impact of Feature Reduction for Deep Learning-based Seasonal Sea Ice Forecasting. In Northern Lights Deep Learning Conference 2025.

Thank you for your comment. Following line 368 in the Discussion section, we have added the following to link our feature importance results with the study by Uebbing et al. (2025):

"Our feature importance findings can be associated with recent work by Uebbing et al. (2025) investigating the impact of feature reduction on seasonal Arctic sea ice forecasting by using the state-of-the-art IceNet model (Andersson et al., 2021) combined with explainable AI (XAI) techniques. Their study showed that using only a subset of key features (such as historical sea ice concentration, linear trend forecasts, and seasonal encoding), high predictive accuracy under general scenarios was still obtained. However, their research also highlighted that for extreme events, such as anomalous sea ice extents, models incorporating additional climate variables perform better. This suggests that further studies might benefit from exploring different XAI methods for estimating feature importance and investigating the extent to which the reduction of the number of features affects deep learning model predictions for Antarctic sea ice.

Discussion: Why does the performance differ between the different regions of the Southern Ocean? For example, the disparities between 4b1-f1. There is mention of this on lines 367-369, but please expand further. Also, please comment on the better predictive performance of the tool in the Austral summer.

Thank you for your comment. The differences in model performance across regions could be attributed to regional variability due to oceanographic conditions, sea ice dynamics, and the influence of atmospheric and oceanic circulation patterns. We expanded on this in discussion section. As observed in our results, ANTSIC-UNet shows better predictive performance relative to the two benchmark models and SEAS5 during the Austral summer, particularly in the sea ice edge zone. We added the further discussion of the spatial forecasting performance as follows:

"Our findings are consistent with those of Marchi et al. (2019) and Bushuk et al. (2021) that sea ice concentration prediction tends to be more accurate in the winter months but less so in the summer due to rapid and irregular changes in the ice edge during that season. Inspiringly, ANTSIC-UNet shows lower summer sea ice edge error and SIC RMSE compared to both the two benchmark models and SEAS5, especially during extreme years. The differences in model performance across regions could be attributed to regional variability due to oceanographic conditions, sea ice dynamics, and the influence of atmospheric and oceanic circulation patterns. Regional seas in the West Antarctic, including the Ross Sea, Amundsen Sea, Bellingshausen Sea, and Weddell Sea, exhibit larger interannual variability in sea ice concentration compared to the East Antarctic (Cavalieri and Parkinson, 2008). These regions are influenced by the Circumpolar Deep Water (CDW), with warm-shelf regions such as the Amundsen and Bellingshausen Seas being particularly sensitive to climate changes, with sea ice concentration and the position of the ice edge strongly driven by wind forcing (Stammerjohn et al., 2003; Saenz et al., 2023). The ice flux driven by wind in the Weddell Sea along the Antarctic Peninsula and the Pacific Ocean plays a crucial role in modulating sea ice dynamics, with the dynamical influence being more pronounced in the Pacific sector (Holland and Kwok, 2012). The sea ice increase (decrease) in the Ross Sea (Bellingshausen Sea) is linked to the Amundsen Sea Low (ASL) which is a key climate feature of these regions (Hosking et al., 2013; Turner et al., 2016). In contrast to other regions of Antarctica, sea ice expansion in the Indian Ocean sector is significant throughout all seasons and is associated with surface cooling and ocean renewal processes that stabilize the ocean and limit the intrusion of warmer subsurface waters into the surface layer (Bintanja et al., 2013; Purich et al., 2018). Additionally, seasonal variability in sea ice in the Indian Ocean sector is closely linked to the Southern Annular Mode (SAM) (Yadav et al., 2022)."

**Reference:**

Bintanja, R., van Oldenborgh, G. J., Drijfhout, S. S., Wouters, B., and Katsman, C. A.: Important role for ocean warming and increased ice-shelf melt in Antarctic sea-ice expansion, Nature Geosci, 6, 376–379, https://doi.org/10.1038/ngeo1767, 2013.

Cavalieri, D. J. and Parkinson, C. L.: Antarctic sea ice variability and trends, 1979–2006, Journal of Geophysical Research: Oceans, 113, https://doi.org/10.1029/2007JC004564, 2008.

Marchi, S., Fichefet, T., Goosse, H., Zunz, V., Tietsche, S., Day, J. J., and Hawkins, E.:

Reemergence of Antarctic sea ice predictability and its link to deep ocean mixing in global climate models, Climate Dynamics, 52, 2775–2797, https://doi.org/10.1007/s00382-018-4292-2, 2019.

Holland, P. R. and Kwok, R.: Wind-driven trends in Antarctic sea-ice drift, Nature Geosci, 5, 872–875, https://doi.org/10.1038/ngeo1627, 2012.

Hosking, J. S., Orr, A., Marshall, G. J., Turner, J., and Phillips, T.: The Influence of the Amundsen–Bellingshausen Seas Low on the Climate of West Antarctica and Its Representation in Coupled Climate Model Simulations, Journal of Climate, 26, 6633–6648, https://doi.org/10.1175/JCLI-D-12-00813.1, 2013.

Marchi, S., Fichefet, T., Goosse, H., Zunz, V., Tietsche, S., Day, J. J., and Hawkins, E.: Reemergence of Antarctic sea ice predictability and its link to deep ocean mixing in global climate models, Climate Dynamics, 52, 2775–2797, https://doi.org/10.1007/s00382-018-4292-2, 2019.

Purich, A., England, M. H., Cai, W., Sullivan, A., and Durack, P. J.: Impacts of Broad-Scale Surface Freshening of the Southern Ocean in a Coupled Climate Model, Journal of Climate, 31, 2613–2632, https://doi.org/10.1175/JCLI-D-17-0092.1, 2018.

Saenz, B. T., McKee, D. C., Doney, S. C., Martinson, D. G., & Stammerjohn, S. E. (2023). Influence of seasonally varying sea-ice concentration and subsurface ocean heat on sea-ice thickness and sea-ice seasonality for a 'warm-shelf' region in Antarctica. Journal of Glaciology, 69(277), 1466–1482. https://doi.org/10.1017/jog.2023.36

Stammerjohn, S. E., Drinkwater, M. R., Smith, R. C., and Liu, X.: Ice-atmosphere interactions during sea-ice advance and retreat in the western Antarctic Peninsula region, Journal of Geophysical Research: Oceans, 108, https://doi.org/10.1029/2002JC001543, 2003.

Turner, J., Hosking, J. S., Marshall, G. J., Phillips, T., and Bracegirdle, T. J.: Antarctic sea ice increase consistent with intrinsic variability of the Amundsen Sea Low, Clim Dyn, 46, 2391–2402, https://doi.org/10.1007/s00382-015-2708-9, 2016.

Yadav, J., Kumar, A., Srivastava, A., and Mohan, R.: Sea ice variability and trends in the Indian Ocean sector of Antarctica: Interaction with ENSO and SAM, Environmental Research, 212, 113481, https://doi.org/10.1016/j.envres.2022.113481, 2022.

There is no conclusion section. Please check if this is required.

Thank you for your comment. In the manuscript, we have included a section titled "Discussion and Conclusion", which combines the interpretation of findings with the main conclusion of the study.

---

## Author Response (AR4)

**Response to comments by Reviewer #1**

This reviewer recommended acceptance of the manuscript as is.

We sincerely thank the reviewer for the positive assessment and support of our work.

**Response to comments by Reviewer #2**

We would like to thank the reviewer for the helpful comments on the paper. Please find below our responses to the comments.

Thank you to the authors for addressing my previous comments to this paper, I now believe this manuscript can be submitted subject to a couple of technical corrections:

Figure 4 caption- change Bellingshausen Seas to sea.

Thank you for your comment. We have corrected Bellingshausen Seas to Bellingshausen Sea as suggested.

Thank you for your justification in using a UNet in this study. Please add a line or two in the discussion and conclusion section to reflect that an important future bit of work is comparing model forecast performance across different Antarctic regions and during extreme events using other forms of ML, including generative models.

Thank you for your comment. As recommended, we have added statements in the Discussion and Conclusion section as follows:

"An important direction for future work will be to systematically compare model forecast performance across different Antarctic regions and during extreme events, using alternative ML approaches, including generative models."